



# A Global Daily High Spatial-temporal Coverage Merged Tropospheric NO₂ dataset (HSTCM-NO₂) from 2007 to 2022 based on OMI and GOME-2

Kai Qin[1], Hongrui Gao[1], Xuancen Liu[1], Qin He[1], Jason Blake Cohen[1*]

[1]School of Environment and Spatial Informatics, China University of Mining and Technology, Xuzhou, 221116, China

*Correspondence to*: Jason Blake Cohen (jasonbc@alum.mit.edu)

**Abstract.** Remote sensing based on satellites can provide long-term, consistent, and global coverage of NO₂ (an important atmospheric air pollutant) as well as other trace gases. However, satellite data often miss data due to factors including but not limited to clouds, surface features, and aerosols. Moreover, one of the longest continuous observational platforms of NO₂ observations from space, OMI, has suffered from missing data over certain rows since 2007, significantly reducing spatial coverage. This work uses the OMI based OMNO2 product, as well as an NO₂ product from GOME-2 in combination with machine learning (XGBoost) and spatial interpolation (DINEOF) method to produce a 16-year global daily high spatial-temporal coverage merged tropospheric NO₂ dataset (HSTCM-NO₂, https://doi.org/10.5281/zenodo.10968462, Qin et al., 2024), which increases the global spatial coverage of NO₂ by ~60% compared to the original OMINO2 data. The HSTCM-NO₂ dataset is validated using upward looking observations of NO₂ (MAX-DOAS), other satellites (TROPOMI), and reanalysis products. The comparisons show that HSTCM-NO₂ maintains a good correlation with the magnitude of other observational datasets, except for under heavily polluted conditions ($>6\times10^{15}$ molec.cm$^{-2}$). This work also introduces a new validation technique to validate coherent spatial and temporal signals (EOF) and validates that the HSTCM-NO₂ are not only consistent with the original OMNO2 data, but in some parts of the world effectively fill in missing gaps and yield a superior result when analyzing long-range atmospheric transport of NO₂. The few differences are also reported to be related to areas in which the original OMNO2 signal was very low, which has been shown elsewhere, but not from this perspective, further validating that applying a minimum cutoff to retrieved NO₂ data is essential. The reconstructed data product can effectively extend the utilization value of the original OMNO2 data, and the data quality of HSTCM-NO₂ can meet the needs of scientific research.

## 1 Introduction

The sum of nitrogen dioxide (NO₂) and nitrogen oxide, hereafter referred to as nitrogen oxides (NO$_x$) play several important roles in tropospheric chemistry (Eriksson, 1952; Levy, 1972; Crutzen, 1973; Fishman et al., 1979; Crutzen, 1979; Logan et al., 1981), specifically with respect to tropospheric ozone (Sillman et al., 1990), nitrate aerosol (Lu, Liu, et al., 2021), which indirectly influences radiative forcing both through scattering downward propagating visible light (Richter et al., 2005), as well as through enhancing absorption of black carbon aerosols (Tiwari et al., 2023), and the concentration of tropospheric OH, which indirectly influences both methane and carbon monoxide (Lu and Khalil, 1993; Spivakovsky et al., 2000). During the daytime, under low pollution and low cloud conditions, the photochemical cycle of NO$_x$ can be scaled somewhat stably to NO₂, allowing observations of NO₂ to be an indicator of NO$_x$ concentration (D. Schaub et al., 2006). Under more heavily polluted conditions, such a relationship can also be established, although it is found to vary in space and month-by-month (Qin et al., 2023; Li et al., 2023). Due to its rapid reactivity with water vapor, NO$_x$ forms into nitric acid, contributes directly to acid rain (Wang et al., 2024). Additionally, NO$_x$ has been shown to have adverse effects on human health (Liu et al., 2016), specifically, as an irritant of the respiratory system and via impacts on respiratory diseases when inhaled at high levels (Manisalidis et al., 2020).



The Differential Optical Absorption Spectroscopy (DOAS) method is used extensively to retrieve total column amounts of
trace gases such as $NO_2$ and others based on UV-visible measurements of satellite spectrometers (Eskes and Boersma, 2003).
The DOAS technique is based on the wavelength dependent absorption of light over a specified light path, and it leads to the
application of continuous monitoring of tropospheric pollution levels from space (Platt and Stutz, 2008). Initially applied to
ground-based upward-looking instruments (i.e. MAXDOAS, Wagner et al., 2004), nowadays, satellite-based measurements
have been proven to offer reliable inversions of column $NO_2$ when compared with ground-based measurements (Bauer et al.,
2012; Wang et al., 2017; Ialongo et al., 2020), with the errors commonly within a 20% bound and nearly always within a 40%
bound (Boersma et al., 2004); Irie et al., 2012; Wang et al., 2017; Compernolle et al., 2020; Pinardi et al., 2020; Wang et al.,
2020; Verhoelst et al., 2021).

Satellite observations offer advantages of wide spatial and long-term temporal coverage (Streets et al., 2013), which can help
fill spatial gaps between ground-based observations, and do so using a single platform without the need for calibrating multiple
individual machines (Kolle et al., 2021). Starting nearly two decades ago, and continuing today, an array of different satellites
has been monitoring global tropospheric $NO_2$ distributions including GOME (from 1995 to 2003) aboard ERS-2,
SCIAMACHY (from 2002 to 2012) aboard Envisat, OMI (from 2004) aboard EOS-AURA, GOME-2 (from 2006) aboard
Metop and TROPOMI (from 2017) aboard Sentinel-5P (Richter and Burrows, 2002; Bovensmann et al., 1999; Laan et al.,
2001; Munro et al., 2016; Veefkind et al., 2012). As a result, there have been useful products relating to estimating surface or
near-surface $NO_2$ emissions (Wang et al., 2012; Li et al., 2021) and detecting the long-term or short-term change of $NO_2$ (van
der A et al., 2006; Cooper et al., 2022).

$NO_x$ is emitted any time there is a high temperature reaction that occurs with air present (Echterhof and Pfeifer, 2012). For this
reason, most sources are related to anthropogenic combustion of fossil fuels, biomass, and even forests, as well as a small
amount from natural sources induced by lightning. (Sun et al., 2018; Li et al., 2022; Lu et al., 2021a). Emissions are frequently
computed using a bottom-up approach, where economic, population, and other factors are merged with an activity coefficient
associated with each activity, and applied on average over space and time (Li et al., 2017, Xu et al., 2023). Recent work has
looked at using the satellite observations of $NO_2$ above and applying them on a grid-by-grid and day-by-day basis to attribute
emissions to different types of industrial sources, population centers, power generation, transportation, and residential uses,
agriculture, and natural sources (Li et al., 2023, Qin et al., 2023). Current best estimates vary by considerable amounts from
each other in space and time (Wang et al., 2021), and account for both natural (Deng et al., 2021) and human-based factors
(according to EDGAR and MEIC). There is controversy about the amounts that lightning and microbial activity may or may
not contribute (Logan, 1983).

Vertical column densities (VCDs) of tropospheric $NO_2$ retrieved from satellite-based instruments provide plentiful data under
relatively clean and clear atmospheric conditions, but have many missing pixels in both time and space due to a variety of
factors including very bright surfaces, clouds, and aerosols (Xia and Jia, 2022; Lin et al., 2014). One of the underlying sources
of error is related to the air mass factor (AMF), which allows conversion from a slant column to a vertical column, which is
highly sensitive to cloud and aerosol layer height (Leitão et al., 2010), aerosol absorption (Lin et al., 2014; Cooper et al.,
2019), and the spatial and temporal distribution of NOx emissions (Qin et al., 2023; Li et al., 2023), which can lead to both
uncertainties and biases in the retrieval (Bousserez, 2014). For these reasons, pixels known to be impacted by clouds are
usually filtered before analysis, however other impacted pixels may not be properly filtered, leading to other issues. Similarly,
for some older satellites, due to the orbit and swath width, it respectively requires 3 days, 6 days and 1.5 days for GOME,
SCIAMACHY and GOME-2 to cover the whole globe, additional missing pixels on a day-by-day basis. OMI, which is carried
on a near-polar, sun-synchronous satellite, is the world's first sensor with daily global coverage of $NO_2$ since 2004. However,
since 2007, a reduction in OMI's spatial coverage occurred due to an equipment malfunction, called the row anomaly, which
has slowly grown from two rows in June 2007 to about 50% of the sensor in 2018 (Torres et al., 2018). The absence of data
presently affects both short-term estimation of air quality as well as long-term quantitative analysis (Duncan et al., 2013; van





Geffen et al., 2020), although still is useful for detection of extreme events (Wang et al., 2020; Wang et al., 2021; Deng et al., 2021). Due to 19 years of continuous observations, OMI is a very widely used sensor in the field of atmospheric trace gas research, and finding ways to comprehensively and reasonably fill these missing pixels (He et al., 2020) would allow its
usefulness to be extended into other fields (Wei et al., 2022).

There are many existing approaches to fill missing data from satellite-based platforms including interpolation techniques: geostatistical (e.g., kriging), deterministic (e.g., inverse distance weighted, thin plate splines), and hybrid (e.g., regression kriging) methods (Abdulmanov et al., 2021; Achite et al., 2024), as well as machine-learning techniques include random forests (Sanabria et al., 2013). As there is strong correlation in terms of both geospatial relationships as well as retrieval approaches
used to determine the VCDs between tropospheric $NO_2$ obtained by different sensors (Wang et al., 2016; Park et al., 2020), issues of spatial-temporal correlation need to be carefully taken into consideration, something that these previous approaches may not have fully considered. In this work, the machine-learning and Data Interpolating Empirical Orthogonal Functions (DINEOF) methods are selected to carry out the reconstruction, which take the advantages of both machine learning and pattern recognition in tandem, as demonstrated by previous studies reconstructing satellite chlorophyll-a data (Park et al.,
2020b; Chang et al., 2017; Hilborn and Costa, 2018; Wang and Liu, 2013), filling in missing part of both sea and land surface temperature data (Alvera-Azcárate et al., 2009; Zhou et al., 2017), analyzing sea surface salinity data (Alvera-Azcárate et al., 2016; Chen et al., 2022), and Jiang et al. (2022) used DINEOF to reconstruct the $XCO_2$ data of OCO-2 and OCO-3 by fusing the two effectively improving the spatiotemporal coverage of $XCO_2$ products.

This research aims to accurately and precisely reconstruct the tropospheric $NO_2$ VCD at daily time resolution and grid-by-grid
spatial resolution using OMI 2007-2022. Under the support of global daily High Spatial-temporal Coverage Merged Tropospheric $NO_2$ Dataset (HSTCM-$NO_2$), model validation, spatial distribution analysis and temporal change monitoring can be carried out. Also, HSTCM-$NO_2$ can be an ideal tool for improving numerical prediction of air quality and AMF, contributing to better understanding of typical chemical and dynamic processes in the atmosphere, and future remote sensing retrieval improvements.

**2 Materials and methods**

**Table 1:** Summary of the parameters used in this research.

| Data type | Parameter | Abbreviation | Unit |
|---|---|---|---|
| **OMI** | Daily tropospheric $NO_2$ vertical column concentration | OMI | molec.cm$^{-2}$ |
| **GOME-2** | Daily tropospheric $NO_2$ vertical column concentration | GOME-2_$NO_2$ | molec.cm$^{-2}$ |
| | Daily cloud cover | cloud_fraction | % |
| | Water bodies | wb | - |
| | Evergreen needleleaf vegetation | env | - |

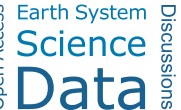

| | | | |
|---|---|---|---|
| | Evergreen broadleaf vegetation | ebv | - |
| | Deciduous needleleaf vegetation | dnv | - |
| | Deciduous broadleaf vegetation | dbv | - |
| | Annual broadleaf vegetation | abv | - |
| **Land cover types** | Annual grass vegetation | agv | - |
| | Non-vegetated lands | nvl | - |
| | Urban and built-up lands | ubl | - |
| | Surface pressure | sp | Pa |
| | Mean surface downward UV radiation flux | msdwuvrf | W/m$^2$ |
| **ERA5 single levels** | Total column ozone | tco3 | kg/m$^2$ |
| | UV visible albedo for diffuse radiation | aluvd | - |
| | UV visible albedo for direct radiation | aluvp | - |
| | Specific rain water content | crwc | kg/kg$^2$ |
| | Ozone mass mixing ratio | o3 | kg/kg$^2$ |
| | Relative humidity | r | % |
| **ERA5 multi-levels** | Temperature | t | K |
| | U-component of wind | u | m/s |
| | V-component of wind | v | m/s |
| | Vertical velocity | w | Pa/s |
| | Latitude | - | - |
| **Others** | Longitude | - | - |
| | Day of year | doy | - |

## 2.1 Tropospheric NO$_2$ products

This work relies on the state-of-the-art Differential Optical Absorption Spectroscopy (DOAS) technique (Platt and Stutz, 2008) to retrieve NO$_2$. This approach uses a DOAS spectral fit of absorption spectra of NO$_2$ and other gases as well as Raman spectra

to a measured reflectance spectrum to determine the slant column density (SCD), which represents the integrated abundance of NO$_2$ molecules along the average photon path through the atmosphere. Next, a calculated AMF is used to convert the SCD into a VCD. Finally, a scheme is applied to separate stratospheric NO$_2$ from tropospheric NO$_2$.

### 2.1.1 OMI tropospheric NO$_2$ (OMNO2)

OMI is a UV/VIS charge coupled device (CCD) spectrometer aboard Aura satellite, which was launched on 15 July 2004 into

a Sun-synchronous orbit with a local equator crossing time of approximately 13:40 h. OMI covers a spectrum of 270–500 nm with a spectral resolution between 0.42 nm and 0.63 nm and a nominal spatial resolution of 13×24 km$^2$ at nadir (Boersma et al., 2008; Foret et al., 2014), providing coverage over 740 wavelength bands along the satellite track and global coverage via



14 orbits per day.

OMI data are processed and archived at the NASA Goddard Earth Sciences Data and Information Services Center (GES DISC). This work specifically uses the daily Level 2G (L2G) gridded data product that corresponds to the OMI $NO_2$ standard product (OMNO2). The L2G products contain one day's worth of L2 data (typically 14 orbits) ordered by ground position rather than by time. Only the most relevant L2 fields are included in the L2G. In an L2G dataset, and the adopted L2G grid is a 0.25-degree by 0.25-degree grid in longitude and latitude.

### 2.1.2 GOME-2 tropospheric $NO_2$

The Global Ozone Monitoring Experiment-2 (GOME-2) is an optical spectrometer aboard the MetOp satellites. MetOp-A was launched on 19 October 2006, MetOp-B was launched on 17 September 2012, and MetOp-C was launched on 7 November 2018. GOME-2 senses backscattered and reflected radiance in the ultraviolet and visible part of the spectrum from 240 nm-790 nm, with a high spectral resolution between 0.26 nm and 0.51 nm covering 4096 spectral points from four detector channels (Fioletov et al., 2013). The spatial resolution varies from 5 km×40 km to 80 km×40 km, and provides daily near global coverage at the equator (Liu et al., 2019).

The GOME Data Processor version 4.8 is used for MetOp-A and -B, while version 4.9 is used for -C.

### 2.1.3 TROPOMI tropospheric $NO_2$

The Tropospheric Monitoring Instrument (TROPOMI) launched on 13 October 2017 aboard the polar-orbiting Sentinel-5 Precursor satellite to measure solar radiation reflected by and radiated from Earth, and provides measurements of atmospheric trace including $NO_2$, $O_3$, $SO_2$, HCHO, $CH_4$, and CO, as well as cloud and aerosol properties. $NO_2$ retrieval is performed using the visible band (400–496 nm), which has spectral resolution and sampling of 0.54 and 0.20 nm. The instrument operates in a push-broom configuration with a swath width of approximately 2,600 km, yielding on Earth's surface. The typical pixel size (near nadir) for $NO_2$ of $7×3.5$ $km^2$ which was reduced to $5.5×3.5$ $km^2$ in 2019 (Ialongo et al., 2020; Ludewig et al., 2020). This work specifically uses the level 2 $NO_2$ data products based on version 1.4, and an applied quality filter of qa_value>0.75 (van Geffen et al., 2019).

### 2.2 Auxiliary data

### 2.2.1 Land cover type data

The Moderate Resolution Imaging Spectroradiometer (MODIS) land cover type (MCD12Q1) provides data that maps global land cover at 500-meter spatial resolution annually for six different land cover legends. The maps were created from classifications of spectra-temporal features derived using the BIOME-Biogeochemical Cycles approach described by Running et al. (1993).

### 2.2.2 MAX-DOAS data

Multi Axis Differential Optical Absorption Spectroscopy (MAX-DOAS) is a passive DOAS ground-based remote sensing observation technology using solar scattering as the light source. MAX-DOAS technology can be used to detect trace gases in the troposphere and has been widely applied in related fields. This instrument can observe scattered sunlight from different perspectives, thus having high sensitivity to trace gases in the troposphere, specifically using low elevation observations as the measurement intensity and zenith measurements as the reference intensity, the Lambert Beer Law can be used to determine the total molecular amount of specific gas categories along the optical path (subtracting zenith concentration from non-zenith measurements), which is known as differential slant column concentration. The tropospheric vertical column concentration is inverted using a radiative transfer model. This work, specifically adopts the QA4ECV $NO_2$ MAX-DOAS reference datasets,



which includes 10 sites, and uses 3 of them. The information of the 3 sites is listed in Table 3.

**Table 2:** Information of MAX-DOAS sites.

| Station | Latitude | Longitude | Range of NO$_2$ Observations (molec.cm$^{-2}$) | Time Zone | Data Used |
|---------|----------|-----------|-----------------------------------|-----------|-----------|
| Uccle (BE) | 50.8°N | 4.4°E | 0-26×10$^{15}$ | 0 | 2011.04-2015.06 |
| OHP (FR) | 43.9°N | 5.7°E | 0-7×10$^{15}$ | 0 | 2007.01-2016.12 |
| Xianghe (CHN) | 39.8°N | 117.0°E | 0-59×10$^{15}$ | UTC+8 | 2010.04-2017.01 |

### 2.2.3 Reanalysis meteorological data

Reanalysis combines model data with observations from across the world into a globally complete and consistent dataset using a model of the atmosphere based on the laws of physics and chemistry. For this reason, this work uses the fifth generation ECMWF reanalysis (ERA5) for 12 specific meteorological parameters as given in Table 1. The dataset used has an hourly temporal resolution and a 0.25° × 0.25° spatial resolution. Those meteorological products in this work are used at the following pressure levels: 100 hPa, 200 hPa, 500 hPa, 700 hPa, 850 hPa, 925 hPa, and 1000 hPa. The actual weightings used in this work are computed using PCA with this data following (Cohen, 2014), and rely nearly fully on the data from 850hPa and below, at roughly equal weights.

This study also uses the fourth generation ECMWF reanalysis (EAC4) specifically for its modeled NO$_2$ column values, which are used as a means of comparison against the NO$_2$ fields generated within this work. The spatial and temporal distribution and vertical levels are similar with the ERA5 data, and a vertical column density is used for comparison in this work.

### 2.3 EXtreme Gradient Boosting algorithm

A gradient boosting framework is used by the decision-tree based ensemble Machine Learning approach known as XGBoost (eXtreme Gradient Boosting, Chen and Guestrin, 2016), because it employs a more regularized model formalization than other techniques (Cisty and Soldanova, 2018; Zhang et al., 2018). For this reason, it has greater control against overfitting compared with gradient boosting decision tree (GBDT) approaches (Dong et al., 2022). Similar to the random forest, the XGBoost needs its hyperparameters tuned (Kapoor and Perrone, 2021). It has a more intricate structure and adds regularization components to the loss function to prevent overfitting so that it can handle complicated data better. Therefore, XGBoost is a better option for working with vast volumes of data and multidimensional affecting factors like NO$_2$ gap filling. Additionally, XGBoost has been used to estimate pollutants, and its results outperform those of certain other statistical and machine learning methods (Reid et al., 2015; Just et al., 2018; Zhai and Chen, 2018; Fan et al., 2020). Table 1 shows the data used in this research, which are input into the machine learning model.

### 2.4 DINEOF method

DINEOF is used in this work to reconstruct the missing points in the spatio-temporal field of NO$_2$. This method relies on an empirical orthogonal function decomposition (EOF) in space and a Principle Component decomposition (PCA) in time that identifies spatial-temporal domains of maximal variation following (Cohen, 2014). The method allows assignment of a prediction under conditions in time and/or space which are missing observational data. By using the weighted EOFs and PCs in an iterative manner, missing data points can be re-synthesized based on a weighting of the various underlying orthogonal basis functions. The number of iterations which minimizes the cross-validation error is used to obtain the best reconstructed data. For a more detailed description of the overall approach, see Beckers and Rixen (2003) and Alvera-Azcarate et al. (2005). In this work, the amount of data filled using this approach ranges from 27% to 35% on a year-by-year basis, as given in Table 4.





**2.5 Statistical indicators**

The root mean square error (RMSE), Pearson correlation coefficient (R), normalized mean bias (NMB) and mean absolute error (MAE) are all used in this work to analyze various aspects of the machine learning model's performance. RMSE (equation 1) is a measure of the deviation between the model prediction $\hat{y}_i$ and the actual OMI observation $y_i$, and is sensitive to outliers, with the smaller the value, the smaller the model error.

$$RMSE = \sqrt{\frac{1}{n}\sum_{i=1}^{n}(y_i - \hat{y}_i)^2} \tag{1}$$

R (equation 2) represents the correlation (in space and time) between the model prediction $\hat{y}_i$ and the actual OMI observation $y_i$, and ranges from -1 to 1, where 1 indicates a perfect positive correlation, -1 indicates a perfect negative correlation, and 0 indicates no correlation. In this work, all values of R are considered only when the p value is smaller than 0.05.

$$R = \sqrt{1 - \frac{\sum_{i=1}^{n}(y_i - \hat{y}_i)^2}{\sum_{i=1}^{n}(y_i - \overline{y})^2}} \tag{2}$$

NMB (equation 3) represents the deviation size of the model prediction $\hat{y}_i$ and the actual OMI observation $y_i$. A smaller absolute value of NMB means a smaller deviation, indicating consistency with the model variability.

$$NMB = \frac{\sum_{i=1}^{n}(y_i - \hat{y}_i)}{\sum_{i=1}^{n}y_i} \tag{3}$$

Finally, MAE (equation 4) represents the average absolute value of the difference between the model prediction $\hat{y}_i$ and the actual OMI observation $y_i$; the smaller the value, the better the model reflects the mean or central tendency of the data.

$$MAE = \frac{1}{n}\sum_{i=1}^{n}|y_i - \hat{y}_i| \tag{4}$$

**2.6 Empirical Orthogonal Functions**

**2.7 Method selection**

The goal of this work is to use all available day-by-day and pixel-by-pixel NO₂ column data from both GOME-2 and OMI in tandem to reconstruct a consistent global NO₂ column product with the highest possible coverage. Machine Learning used in this work can only predict OMNO2 data which also has GOME-2 data at corresponding position in space and time. For this reason, this work introduces DINEOF to reconstruct data in locations where both of OMI and GOME-2 do not have values, but where data exists at other times or nearby locations in space.

Since DINEOF and machine learning have not previously been used in tandem for this type of issue, a critical component of the methodology is to quantify the impact of using the two approaches individually, in tandem, and if in tandem in what order.

To first determine which sets of methods are best suited for this work, a subset of data from 2007 is selected. Furthermore, due to the issue of the row anomaly, a second comparison dataset from 2013 is used as a mask. In this way, data from 2007 which are masked by data from 2013 will be separated for validation, and the missing data will be the major difference assuming the changes in the climatology are not significant. Therefore, the following methods are applied, as displayed in Fig. 1:

I.   First XGBoost is used to predict OMNO2 data based on GOME-2 data. Subsequently, DINEOF was applied to fill the remaining gaps.

II.  DINEOF is first used to fill the gaps in GOME-2 data. This is then followed by XGBoost prediction based on the reconstructed GOME-2 dataset.

III. DINEOF is used solely to fill in gaps in OMNO2.

The reconstructed dataset is evaluated based on comparison between the masked data from 2007 and the results. Additionally, in order to verify whether and how the absence of GOME-2 values affects prediction accuracy, further partitioning of the



dataset based on the presence or absence of GOME-2 values is performed. All results are given in Fig. 1, where the row is the method and the column is the amount of GOME-2 data used.

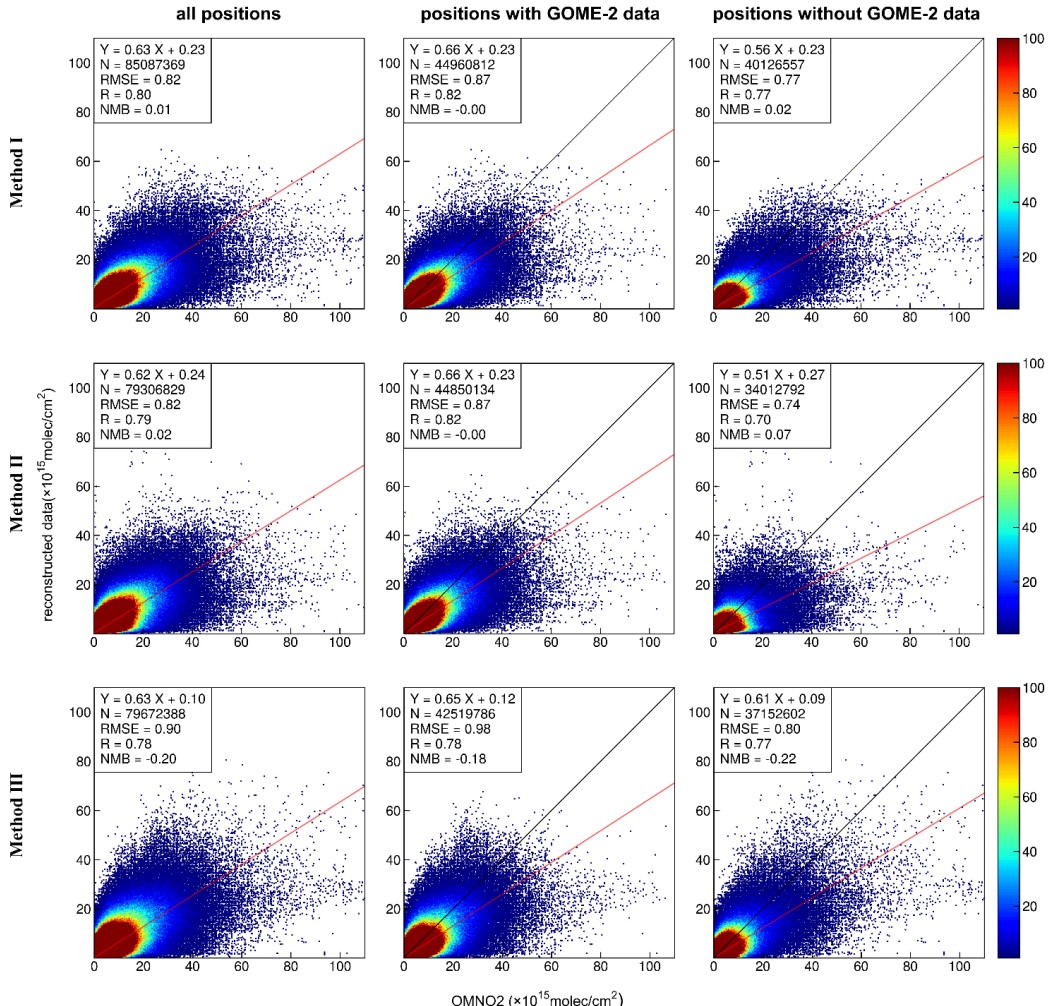

**Figure 1:** Cross-validation of 3 methods between reconstructed data and masked OMNO2 in 2007.

As shown in Fig. 1, the complete or partial datasets reconstructed by Method I all have the maximum R value and the minimum RMSE and NMB in the same scenario. Meanwhile, by comparing Column 2 and Column 3, it's obvious that the presence of GOME-2 observations can greatly improve the accuracy of reconstruction. From Row 3, it can be found that DINEOF has universality, but does not have outstanding performance. Therefore, it is necessary to use machine learning for prediction in positions with values obtained from GOME-2 and DINEOF only used for filling in positions which do not contain GOME-2.

In conclusion, in order to obtain the optimal results, Method I will be chosen as the reconstruction scheme in this work, which is consistent with the idea that using the most about of actual observational data possible best supports the machine learning approach.

## 3 Results

### 3.1 Reconstruction process and model evaluation

### 3.1.1 Quantifying the importance of individual features

The relevance of a feature in machine learning may be determined by contrasting the projected values of the model with and without inclusion of that feature, for example by exclusion. Shapley Additive exPlanations (SHAP) values quantitatively represent the conditional expected value function of the machine learning model, implying the average contribution of a feature to a prediction (Lloyd Shapley, 1952. Use of a black box model, such as XGBoost in this work, necessitates an explanatory model in contrast to interpretable algorithms (i.e. Cohen and Prinn, 2011). According to each feature's marginal contribution, the SHAP distributes the overall gain, in terms of both negative and positive contribution. The results of the SHAP value and its statistics are given in Fig. 2 based on global training data from February 2019 as an example.

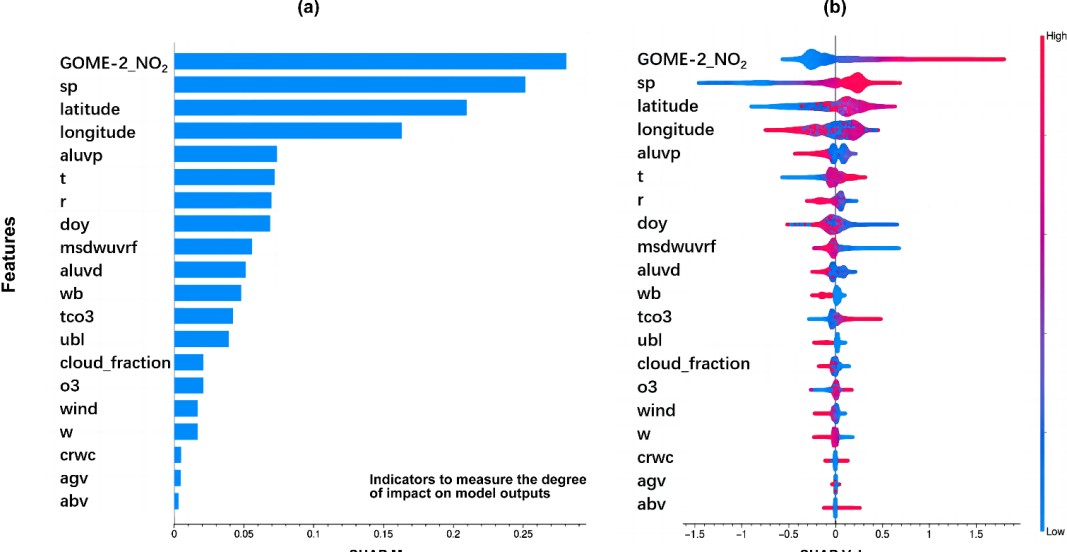

**Figure 2:** Feature importance ranking **(a)**, and Scatter plot of feature density for each parameter of XGBoost **(b)**, represented as a beeswarm. In specific, each row represents a feature, and the order of arrangement is determined by the importance of the feature calculated in the previous step. The horizontal coordinate is the SHAP value, where the sign of the value indicates the direction of contribution of that feature. Each point in each row represents a single sample, and the color of the point indicates the magnitude of the feature value (high in red and low in blue).

The 20 features with the highest contribution are provided. Data from GOME-2_NO$_2$ has both the highest overall mean contribution, as well as the largest absolute contribution (up to 1.8), which is larger than the absolute values of all other contributing factors, as well as the only significant source in terms of positive contribution (greater than 0.6). This result is consistent with the fact that GOME-2_NO$_2$ is the base observation upon which the machine learning is acting. The second most significant driving feature is the surface pressure, which has both the second highest mean and the second largest absolute contribution (down to -1.5) of any factor. This is consistent with the fact that human settlements tend to occur at lower elevations in general, that changes in pressure tend to accompany changes in the rates of transport and chemical activity of NO$_2$ in-situ (Li et al., 2023; Wang et al., 2020). Below this there are some interesting patterns in which some species contribute more to the mean SHAP, but not necessarily to the extreme SHAP values, meaning that the global and local contribution factors are different in different locations. As expected, latitude, longitude, day of year, and downwelling UV radiation are all relatively important in different areas, which is consistent with the highly heterogenous nature of NO$_2$ emissions, different driving forces which impact the ratio of NO to NO$_2$ emissions within NO$_x$, issues of geospatial change, and processing once the NO$_2$ is in the atmosphere, among other factors. These factors are sufficient to capture the presence of pollution sources within specific certain pixels, and therefore it is required to not only be able to predict the long-term signal, but also account for short-term changes of a sudden nature as well.

### 3.1.2 Separation of models over water and land

Globally, the distribution of NO₂ observed by satellites is not balanced, due to the fact that NO₂ has a relatively short lifetime, and the vast majority of its emissions occur over land in and around areas of anthropogenic disturbance. For this reason, excluding significant shipping lanes and significant downwind transport clear to the short, lower general values are expected over the sea. On top of this, the surface absorption profile over the oceans is different from land, which may further contribute to differences in the column interpretation. This section quantitatively explores the impact of separating those pixels over the

sea from those over the land in terms of training the machine learning model, and works to quantify any reduction in the overall error rate of the models between the separated and unified approaches.

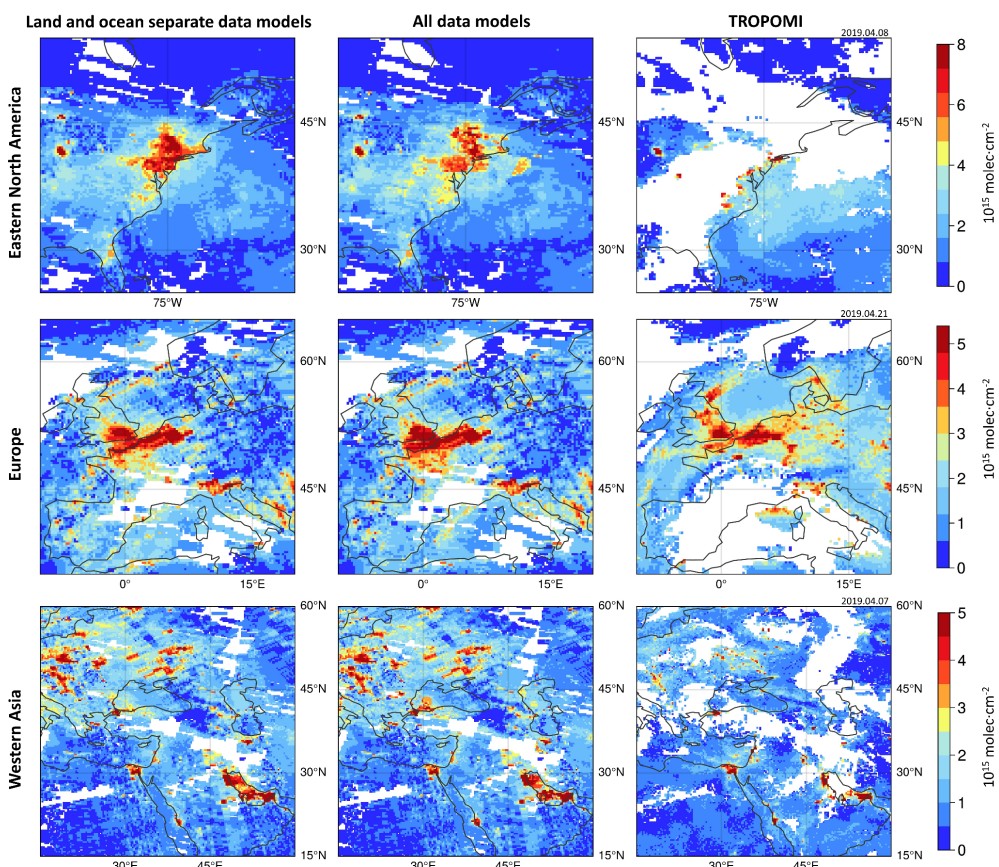

**Figure 3:** Prediction results with and without prior knowledge, versus TROPOMI observations over 3 regions (Eastern North America, Europe and Western Asia).

The effects of separating the land from the ocean models are demonstrated clearly over April 2019 in Fig. 3. First, the high values of NO₂ observed over the Western Atlantic Ocean found in the all data model are no longer observed in the land and ocean separate data models, which is consistent with TROPOMI NO₂ observations. Over western Europe, the high values off of Scotland as confirmed by TROPOMI still remain in the land and ocean sperate data models' case, while the unusually high values in the all data model case are reduced to more reasonable values compared to the observations from TROPOMI over

the areas between Spain and France and between the UK and France. Even with the separation, there are still erroneously high values between the UK and Ireland, and in the Eastern Atlantic which are not resolved. The third row shows the distribution of NO₂ concentrations in western Asia. In the TROPOMI observations, high values are observed on the southwest side of the Arabian Peninsula only over land, and mostly on land over Northern Turkey except for the Bosphorus Straits, which is



consistent with what is understood, and which the separate land and ocean data models are able to capture, while the all data
model misrepresents these values as being higher than the observations support. Overall, there is a considerable improvement
observed over near-sea areas, in terms of both retaining enhancement where it is justified and reducing enhancement where it
is not justified by using the separately trained models. However, there still are inconsistencies which are not resolved.

### 3.1.3 Evaluation of machine-learning

After applying XGBoost and prior knowledge based on the validation dataset mentioned in 3.1, Fig. 4 demonstrates the
reconstructed results and compares them with the original data on a pixel-by-pixel basis in 2007.

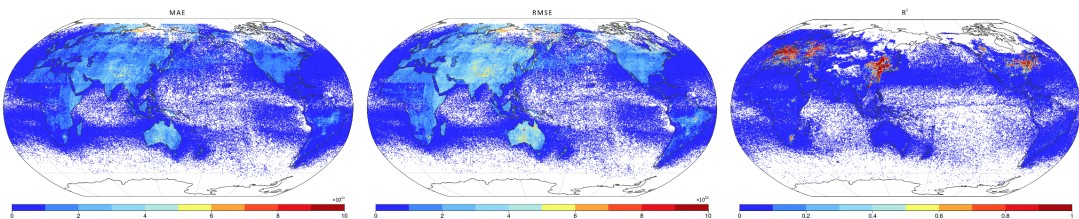

**Figure 4:** Accuracy validation of XGBoost prediction results (MAE, RMSE, and $R^2$).

Among the results predicted by XGBoost, the MAE and RMSE of the results located over water are slightly lower than those
located over land. The predicted results over Eastern Asia, Europe and Eastern North America show a higher correlation with
the observations than the original data, indicating that the variability is captured better over regions where the vertical column
concentration of $NO_2$ is larger. For these reasons, the machine-learning model is trained separately over both land and over the
ocean, with training done on a month-by-month basis. The results of this fitting are given in Fig. 5, demonstrating the time
series of the statistics of the from 2013-2015.

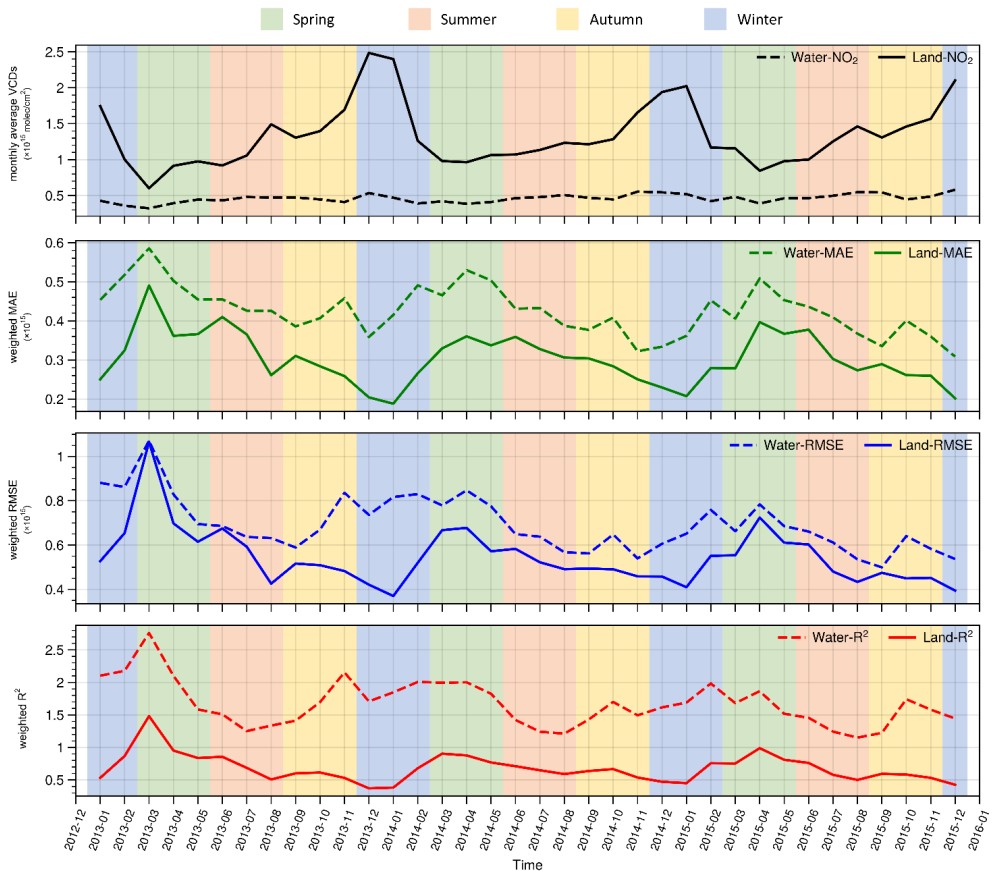

**Figure 5:** Land and water model quality of XGBoost from 2013-2015.

As demonstrated, the RMSE and MAE of the water model are both always higher than, and less temporally variable than those of the land model, without gaining the advantage of improvement observed during the land model peak in the winter. This indicates that these errors scale both in terms of the magnitude, which is higher over land, but also in response to the retrieval algorithms themselves, which have a different amount of error over bright and dark surfaces, as well as in response to surface based measurements of which there are more over land to have initially improved the retrieval precision. , The correlation over time of the land model is slightly higher than that of the water model, indicating that the data predicted by the land model may have a lower uncertainty, possibly due to a better a priori data, a better defined AMF over land, or due to the overall retrieval being better over land as compared to water (Richter et al., 2011, Streets et al., 2013b, Lamsal et al., 2021).

The quality of the land model fit shows a strong decrease in quality over a period of 1 to 3 months every year, experiencing both inter-annual and intra-annual variation, while the ocean model shows a weaker decrease in the fit for a few months in two of the years, and no change in the other years. This indicates that there must be a few different forces acting upon the fits, including some of which are clearly seasonal in nature with only small variation (air temperature), while others are more variable (UV radiation and AAOD), consistent with the results from (Li et al., 2023; Wang et al., 2020). On the one hand, the UV intensity is generally lowest in December and January, leading to an increase in the residence time of $NO_2$ in the atmosphere, and generally highest in May and June, leading to a decrease in the residence time. However, the UV itself also modified by the effects of both clouds and absorbing aerosols. The effects of clouds over the ocean tend to be more in terms of overall percentage of the surface covered than over land, but the effects of absorbing aerosols tend to be more over land, leading to the above findings. The effects of temperature tend to peak differently from those of UV radiation, but these effects



tend to be climatologically more similar year to year, given that the years analyzed do not contain any El Niño or La Niña
types of patterns. In addition to this, the vertical column concentration of $NO_2$ itself also changes from month-to-month with
the peak values over land occurring in December and January, with both the magnitude of the peak and the peak month varying
from year-to-year. This allows for a greater amount of differentiation between the heavily polluted and more clean regions
during this time, especially so over land. As discussed previously, such high variability may lead to additional machine-
learning fitting issues. On the other hand, there is generally less cloud during the winter, meaning more observations on a day-
by-day basis, as well as more atmospheric stability in the winter, leading to less vertical and long-range transport pollutants
away from their source regions. The combination of all of which enable the model to achieve more accurate predictions.

### 3.1.4 Reconstruction process and accuracy analysis of DINEOF

The EOF separates the data into its primary basis functions, of which there are spatial and temporal components. To test the
efficacy of the EOF procedure as a function of the time length of data used, this work has run the procedure over different time
periods from a minimum of 1 month of data to a maximum of 3 years of data. The annual data, as shown in Table 4, yields the
lowest overall standard deviation. This is consistent with the above results showing that there is a clear annual peak in the $NO_2$
columns occurring each winter, and indicates that this amount of variability drives the model more than the smaller year-to-
year changes in the peak or overall characteristics of $NO_2$. This result is consistent with a year (intra-annual variability) tending
to be smaller than year-to-year variability unless a very long time series is considered (minimum of 20-30 years) (Chowdhury,
2022), unless capturing a known extreme such as El Niño, La Niña, etc. (Deng et al., 2021). Based on the timing chosen and
the results below, this work will rely upon applying the DINEOF reconstruction of the dataset on a year-by-year basis.

**Table 4:** Reconstruction results of different time lengths of DINEOF.

| Time Length | 1 month | 3 months | 6 months | 1 year | 3 years |
|---|---|---|---|---|---|
| Start Time | 2008.01.01 | 2008.01.01 | 2008.01.01 | 2008.01.01 | 2008.01.01 |
| End Time | 2008.01.31 | 2008.03.31 | 2008.06.30 | 2008.12.31 | 2010.12.31 |
| Image Number | 31 | 91 | 182 | 366 | 1090 |
| Missing Rate | 34.2% | 27.7% | 29.1% | 29.3% | 32.0% |
| Mean Value ($\times 10^{15}$) | 0.60 | 0.54 | 0.54 | 0.56 | 0.58 |
| Standard Deviation ($\times 10^{15}$) | 1.57 | 1.35 | 1.15 | 1.12 | 1.13 |
| Iterations Made | 112 | 50 | 16 | 12 | 12 |
| Convergence Achieved | 10.0E-4 | 10.0E-4 | 9.8E-4 | 8.61E-3 | 9.9E-4 |

Table 5 shows the DINEOF results for each year, with most years achieving convergence after 12 to 29 iterations. The standard
deviation is shown to be lowest when analyzing data one year at a time. Interestingly, the year with the most lost data was in
2009, in which a significant more than one third of the total data (34.3%) was lost, indicating that the geospatial nature of the
data lost and its overall range of column loading values both matter in addition to the absolute amount of data reconstructed.

**Table 5:** Statistics of DINEOF reconstruction results by year.

| Year | Mean Value ($\times 10^{15}$) | Standard Deviation ($\times 10^{15}$) | Iterations Made | Convergence Achieved | Missing Rate |
|---|---|---|---|---|---|
| 2007 | 0.56 | 1.18 | 12 | 9.2E-04 | 31.1% |
| 2008 | 0.55 | 1.12 | 12 | 8.5E-04 | 29.3% |
| 2009 | 0.58 | 1.07 | 16 | 9.7E-04 | 34.3% |
| 2010 | 0.59 | 1.17 | 16 | 9.4E-04 | 32.5% |
| 2011 | 0.59 | 1.21 | 14 | 8.8E-04 | 33.1% |
| 2012 | 0.59 | 1.21 | 13 | 9.4E-04 | 30.7% |
| 2013 | 0.59 | 1.16 | 12 | 9.1E-04 | 23.0% |
| 2014 | 0.58 | 1.05 | 14 | 9.3E-04 | 23.5% |
| 2015 | 0.58 | 0.98 | 22 | 9.6E-04 | 22.6% |
| 2016 | 0.60 | 0.91 | 18 | 9.9E-04 | 23.9% |





| 2017 | 0.59 | 0.93 | 22 | 9.3E-04 | 32.2% |
| 2018 | 0.58 | 0.92 | 29 | 9.9E-04 | 23.2% |
| 2019 | 0.59 | 0.59 | 25 | 9.9E-04 | 21.2% |
| 2020 | 0.58 | 0.86 | 42 | 9.8E-04 | 21.4% |
| 2021 | 0.64 | 0.93 | 64 | 9.9E-04 | 22.9% |
| 2022 | 0.68 | 0.88 | 23 | 9.6E-04 | 21.3% |

### 3.1.5 Overall analysis

The performance of reconstruction can be tested by simulating the known "line anomaly" issue, wherein OMI started to lose
access to specific camera angles on a swath-by-swatch basis starting in 2007, leading to the appearance of missing lines of
data. Since the data is otherwise in good order, a well-conditioned filling method should be able to produce data to cover these
well-known and geometrically simple gaps. Five regions (East Asia, Europe, North America, Africa, and South Asia) are used
to demonstrate the effectiveness of the procedure to fill these gaps on different days in 2007.

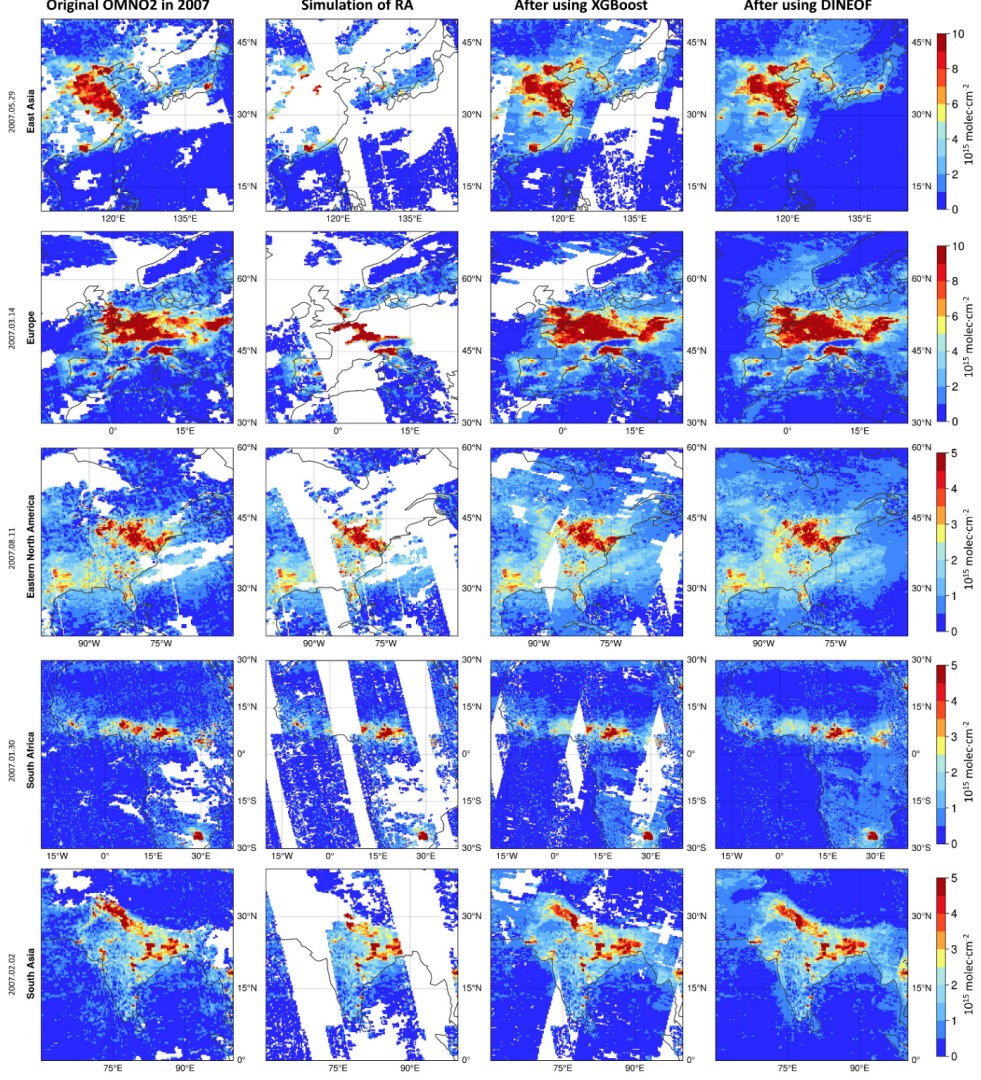

**Figure 6:** Results of stepwise reconstruction of masked data over 5 regions (East Asia, Europe, Eastern North America, South Africa, and
South Asia).

As shown in Fig. 6, the first column shows the original OMNO2, and the second column shows the data distribution after simulating the effect of "line anomaly". The machine learning reconstructs the image of GOME-2 at positions with value, keeps the original observation data, and only reconstructs the missing parts. After reconstruction by XGBoost, the image elements that are still missing are reconstructed using DINEOF to obtain a dataset with more than 99% coverage. Comparing the reconstructed data with the original data, we can find that the reconstructed results are basically consistent with the distribution of the original OMNO2, with the following two exceptions: some very high pixels observed in the EU and USA have been removed and replaced with lower value pixels in the reconstruction, while some moderate and low pixels in China and India have been replaced with high value pixels in the reconstruction. In general, the overall shapes are reasonably similar and the transition from high to low values seems to make sense based on the values from the original OMNO2.

### 3.1.6 Coverage statistics

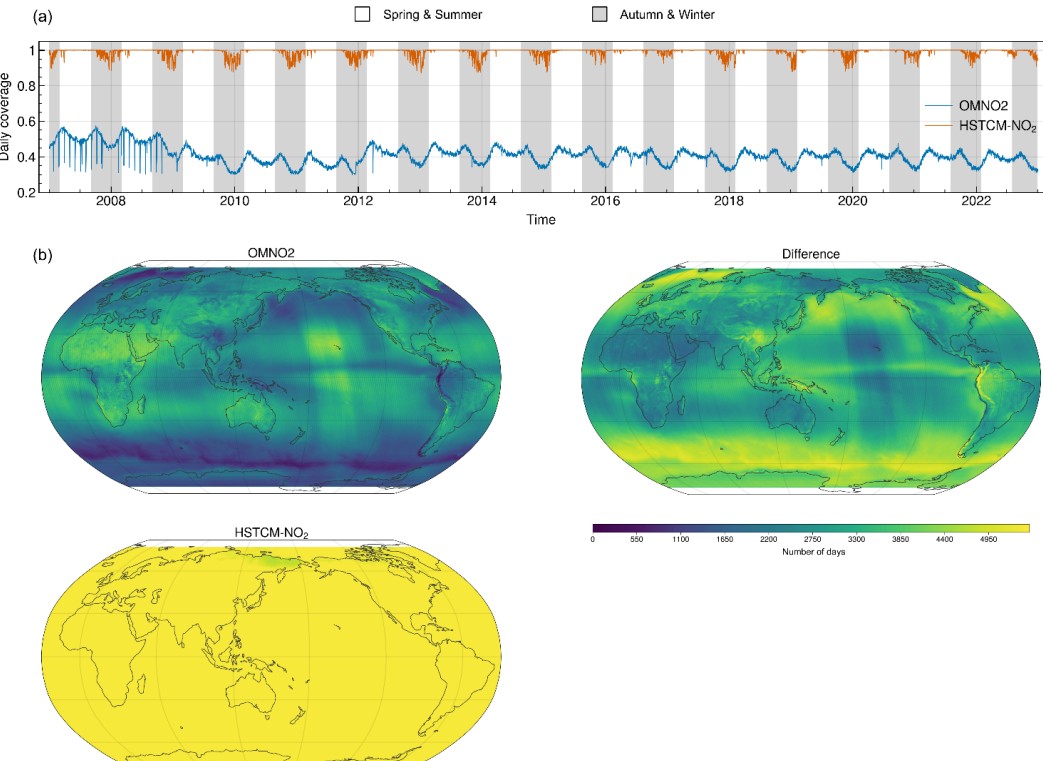

**Figure 7:** Coverage statistics of HSTCM-NO$_2$ from 2007 to 2022, daily coverage (0.3 is used as a cut-off) is shown in **(a)**, number of days with data for each pixel is shown in **(b)**.

The coverage of the original OMNO2 declined from about 50% in 2007 to 35%-40% after 2009 due to the "line anomaly" phenomenon and cloud occlusion, and improved slightly as corrections were applied in late 2012 (although not recovering to previous 2009 levels). The reconstructed data however has a daily coverage of over 90%. As shown in Fig. 7(a), the original data have more gaps when the cloud volume is higher and less data when the cloud volume is smaller. The reconstructed data also shows such a trend, although with a much smaller difference between the high cloud and low cloud periods of time, indicating that some fraction of cloud covered data can be reconstructed successfully, while some other amount has so much data lost that even this technique used in this work cannot fully reconstruct the data.

Fig. 7(b) shows the comparison between the original OMNO2 and the HSTCM-NO$_2$ in terms of spatial distribution. OMNO2 in the eastern part of North America, northwestern part of South America, Europe and southeastern part of Asia are obviously





missing, although after reconstruction all the data in the above locations are reconstructed. The reliability of their HSTCM-NO$_2$ is verified over such land-based and near-land areas. There are a few exceptions, such as perpetually cloud covered areas in the North Pacific and along the equator, but in these cases, there is likely no possible solution since they are covered for days in a row over huge spatial areas. Globally on average, the 39% coverage of OMNO2 increases to a 99% coverage of HSTCM-NO$_2$.

**3.2 Multi-source validation of HSTCM-NO$_2$**

**3.2.1 Comparation with MAX-DOAS data**

The original OMNO2, HSTCM-NO$_2$ were validated against MAX-DOAS and the following results were obtained. The 3 sites used in this work are Xianghe, Uccle and OHP, which are located separately in Sub-urban, Urban and Rural.



**Figure 8:** Scatter plot of comparison between MAX-DOAS observations (Xianghe, Uccle and OHP) and HSTCM-NO$_2$, the figures of the
left panel uses all observations of MAX-DOAS, while of the right panel are filtered out extreme cases.





Comparisons between the various different products and MAX-DOAS are shown in Fig. 10. Due to the small amount of data, there is a missing box which corresponds to a result that didn't pass the p-test. In all cases, there is a sufficient number of data points to consider the fits under both all data and extreme event filtered data conditions. At the site with very high $NO_2$ column loading (Xianghe) and of moderately high $NO_2$ column loading (Uccle), the results using both XGBoost and DINEOF together

are still less good than the original OMI data, regardless of whether the data is filtered or not filtered. In Xianghe this difference is even larger than in Uccle, confirming that the approaches employed here do not work very well when a substantial amount of data is located at or above $6 \times 10^{15}$ molec.$cm^{-2}$. However, it is clearly shown even at these high sites that using both XGBoost and DINEOF together yields a final product which is more representative of OMI than using only one method independently. In the case of Xianghe, using all data with XGBoost alone, or using filtered data with either XGBoost or DINEOF alone yield

similar results, which are worse than applying both XGBoost and DINEOF in tandem. At Uccle, applying XGBoost on its own always yields a result with a much higher R coefficient and a lower RMSE coefficient than when DINEOF is applied on its own, consistent across both filtered and unfiltered data. Interestingly under the relatively cleaner conditions found at OHP, the results of applying both XGBoost and DINEOF together yield a result which is better than the result of OMI in terms of RMSE and similar in terms of R. The application of either XGBoost or DINEOF independently at this location also both yield results

which are quite good when compared with OMI. This set of results makes it clear than under cleaner conditions, the use of one or both of XGBoost and/or DINEOF yield benefits and can be considered trustworthy, while their combination yields a large amount of additional data and still works well. Clearly the benefits of the gap filling and prediction are consistent with the observations under these conditions, allowing conclusions observed above under higher polluted and lower polluted conditions to be further supported.

**3.2.2 Comparation with TROPOMI**

Compared with OMI, TROPOMI has a higher spatial resolution and wider swath angle, allowing improved spatial observation tropospheric $NO_2$, with the caveat that higher resolution may mean that some pixels are cloud-covered, whereas at lower resolution this may not be the case. For these reasons, TROPOMI-$NO_2$ is used as an external data source to allow comparison with the various products and to serve as a means for ensuring that the derived products are reasonable.

The spatial distribution on 4 specific days over East Asia, South Asia, Europe and North America are used to compare OMNO2, HSTCM-$NO_2$ and TROPOMI-$NO_2$, as displayed in Fig. 9.

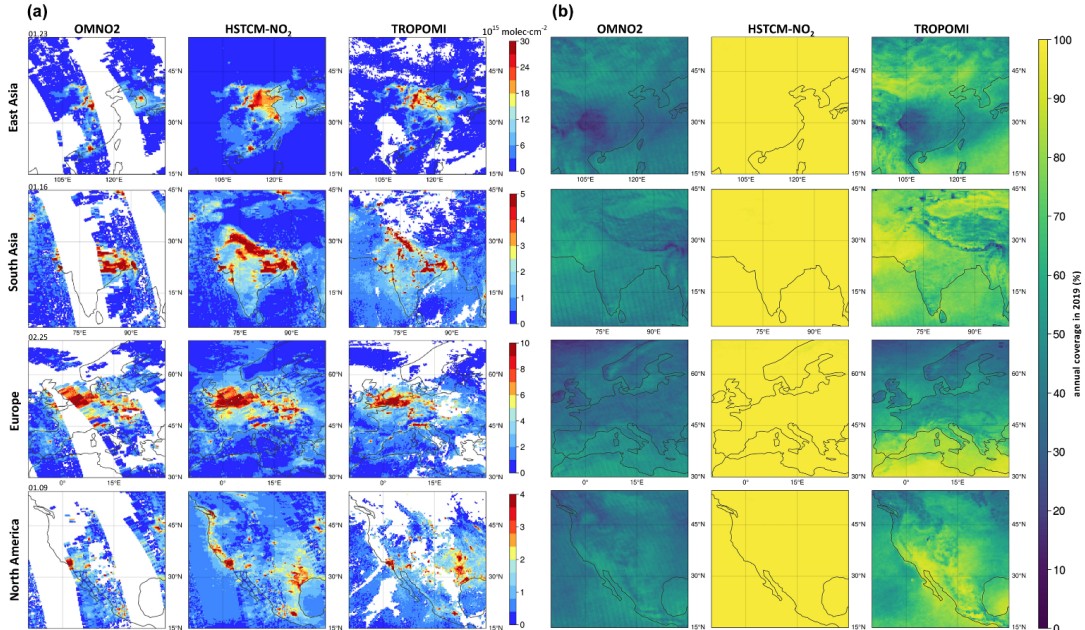

**Figure 9:** Distribution **(a)** and coverage statistics **(b)** of OMI, HSTCM-NO$_2$ and TROPOMI over 4 regions (East Asia, South Asia, Europe and North America).

As shown in Fig. 9(a), in the coastal areas of China is severed affected by the RA, leading to OMNO2 missing a significant portion of data. The reconstructed results of HSTCM-NO$_2$ are similar to TROPOMI on average, and in particular in Hebei, Henan, Shanxi, Shaanxi, parts of industrial Inner Mongolia, the Pearl River Delta, and even the transport corridors between China and South Korea in the East China Sea. However, there are some regions in Shandong, southern Jiangsu, Wuhan, and Shanghai, where the characteristics on average may be acceptable, but where high and low values are too smoothed over and

extremes are not well predicted by HSTCM-NO2 as compared with TROPOMI. Due to the effects of cloud cover, both OMNO2 and TROPOMI show no data over the megacities of Chongqing and Chengdu, while HSTCM-NO$_2$ effectively solves this problem in terms of large-scale spatial averaging, with a coverage of almost 100%. However, the fine-scale centers of the two cities are not clear under this case.

In Fig. 9(b) again due to the RA OMNO2 lacks data over New Delhi, Lahore and other cities in central and western India. The

reconstructed HSTCM-NO$_2$ products fill this part of data well, and the NO$_2$ distribution shown in HSTCM-NO$_2$ is similar to that of TROPOMI, with the major issue being that heavily polluted areas are more diffuse than in TROPOMI. In particular, the areas of Northeast India which are known to have seasonal fires this time of the year are reflected well in HSTCM-NO$_2$ but not in TROPOMI, possibly indicating that the information from the morning provided by GOME-2 identifies information which is missed by TROPOMI in the afternoon. The special geographical environment of the Qinghai-Tibet Plateau has led to

both high cloud cover and significant surface reflection in the region. As a result, the coverage of OMI and TROPOMI products in the Qinghai-Tibet Plateau region is relatively low, and HSTCM-NO$_2$ is able to provide some amount of geospatial information, likely again from GOME-2, while filling the climatological gap.

As shown in Fig. 9(c), due to the influence of the marine climate, high coverage of cloud often occurs in the European region, which causes significant interference to satellite observations. The coverage of both OMNO2 and TROPOMI products in the

European region is relatively low on this day. The HSTCM-NO$_2$ has effectively reconstructed missing data from the UK from Scotland through London, most Central France, and even into Algeria and Tunisia, while greatly increasing data coverage throughout Europe as a whole.

As shown in Fig. 9(d), missing data in areas such as the western coast of North America, Texas, and Oklahoma have been well

reconstructed. Due to the impact of RA, the spatial coverage of OMNO2 is lower than TROPOMI, and the coverage of both

is not ideal in both high latitude and high altitude regions. Through the comparison of the four regions, it can be found that the

HSTCM-NO$_2$ solves this problem, and has high consistency with TROPOMI-NO$_2$. It works particularly well along the West

Coast from San Francisco up through Vancouver, energy producing areas from Texas through New Mexico, and in general

around urban and energy producing areas along the Eastern edge of the Rockies.

As shown in Fig. 10, each day and grid which contains a value of both HSTCM-NO$_2$ and TROPOMI-NO$_2$ are compared in

2019. The comparison consists of a total of 171297320 pixels, and shows a reasonable fit globally with an RMSE of 0.64, R

of 0.75, and NMB of -0.08. As pointed out elsewhere in this work, at values larger than $6\times10^{15}$ molec.cm$^{-2}$, and especially so

at values larger than $20\times10^{15}$ molec.cm$^{-2}$, there are some small differences in the overall shape.

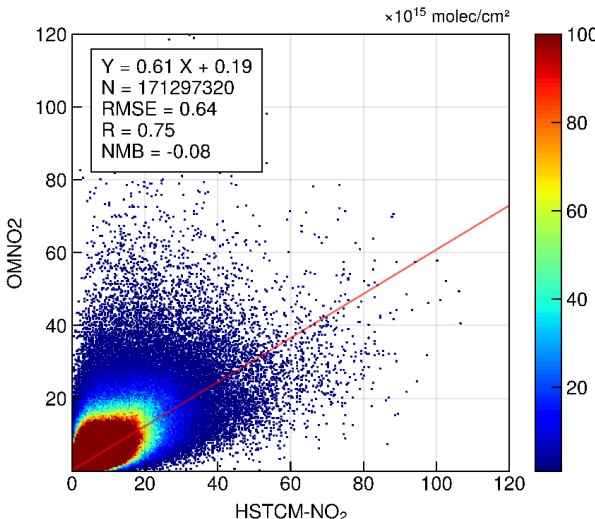

**Figure 10:** Comparison between global HSTCM-NO$_2$ and TROPOMI data in 2019.

**3.2.3 Comparation with EAC4 data**

Global results as well as results over three regions with a sufficient number of pixels with high NO$_2$ vertical column

concentrations (East Asia, North America, and Europe) were selected to compare the reconstruction results with EAC4 data

for February 2008. The reconstructed data at global scale contains a more than 2.6 million points and has an RMSE of 0.9, R

of 0.73, and NMB of 0.3. Among the 3 regions, East Asia has the validation results with the highest R and the lowest NMB,

followed by North America and Europe, as displayed in Fig. 11.

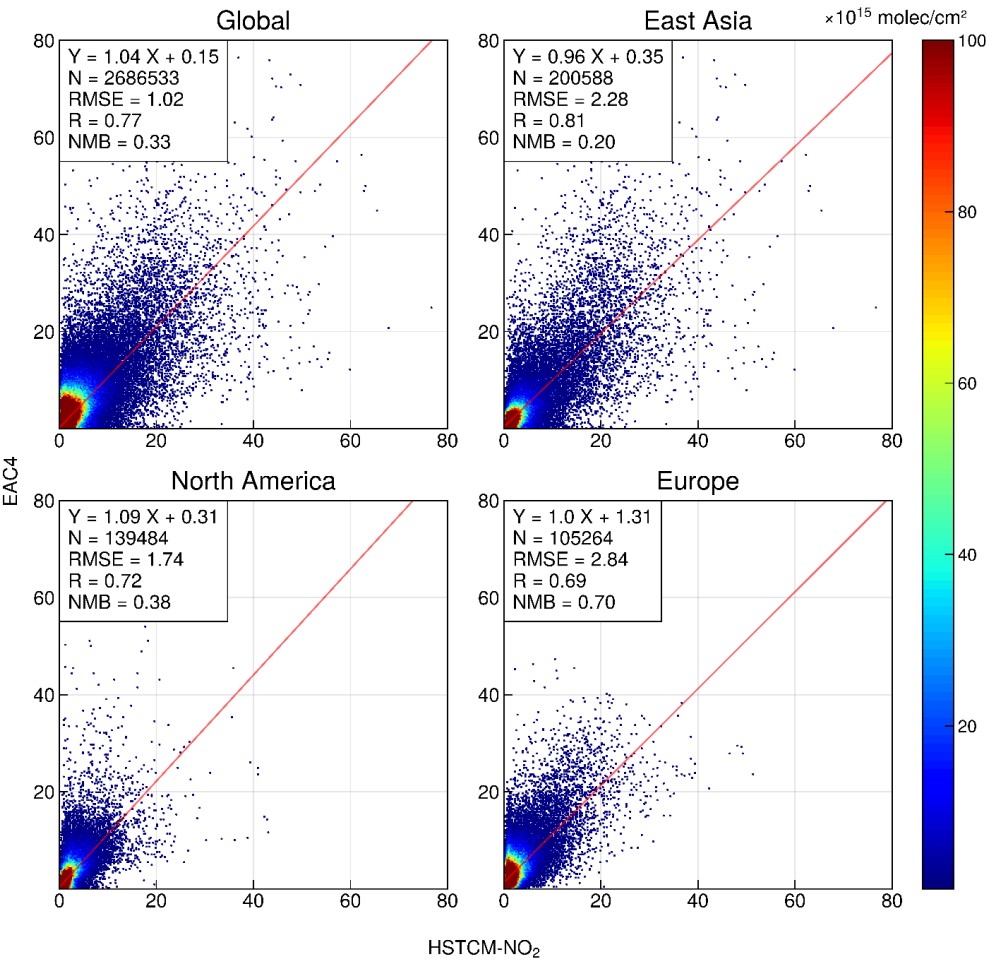

**Figure 11:** Global and regional (East Asia, Europe and North America) comparison of HSTCM-NO$_2$ and EAC4 data.

### 3.3 Results of EOF analysis

In order to verify the performance of HSTCM-NO$_2$, the temporal and spatial patterns are expected to match the observed variability. In specific, analysis was done over the time period from 2019-2021. The first three modes contribute 7.6%, 2.2%, and 2.0% of the total original OMNO2 respectively, while they contribute 26.1%, 4.0%, and 3.2% respectively for HSTCM-NO$_2$. This indicates that a spatial and temporal comparison using the first mode is sufficient to demonstrate the ability of HSTCM-NO$_2$ to reproduce OMNO2, given the fact that they both contribute more than the global background 5% or so error associated with the NO$_2$ retrieval itself. The contribution of HSTCM-NO$_2$'s first mode to the total variance indicates that the reconstructed data is missing many finer modes of variability, however, as demonstrated below, the good spatial and temporal match shows that it is able to reproduce the signal reasonably well in actuality, with the major sources of this difference being regions north of 60°N and south of 60°S, both of which tend to be relatively clean and have the majority of their variability due to noise in the retrievals themselves, which is not explicitly considered by the methods employed herein.



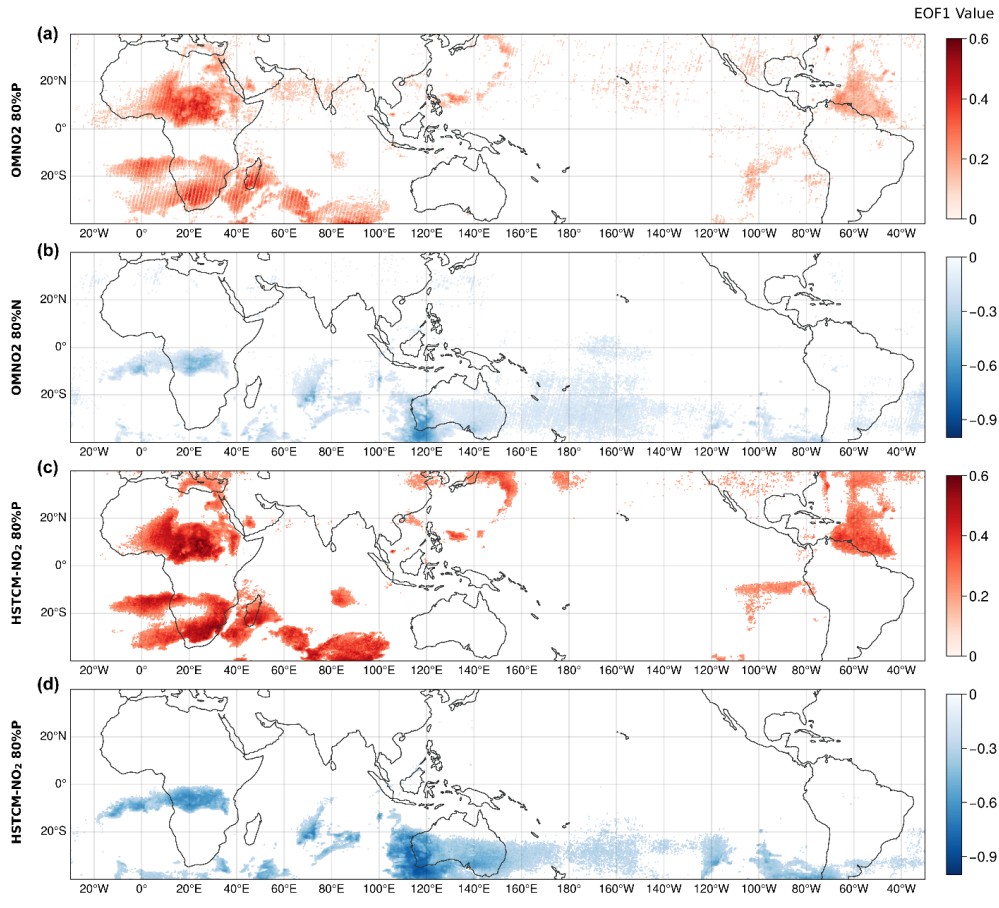

Earth System
Science
Data

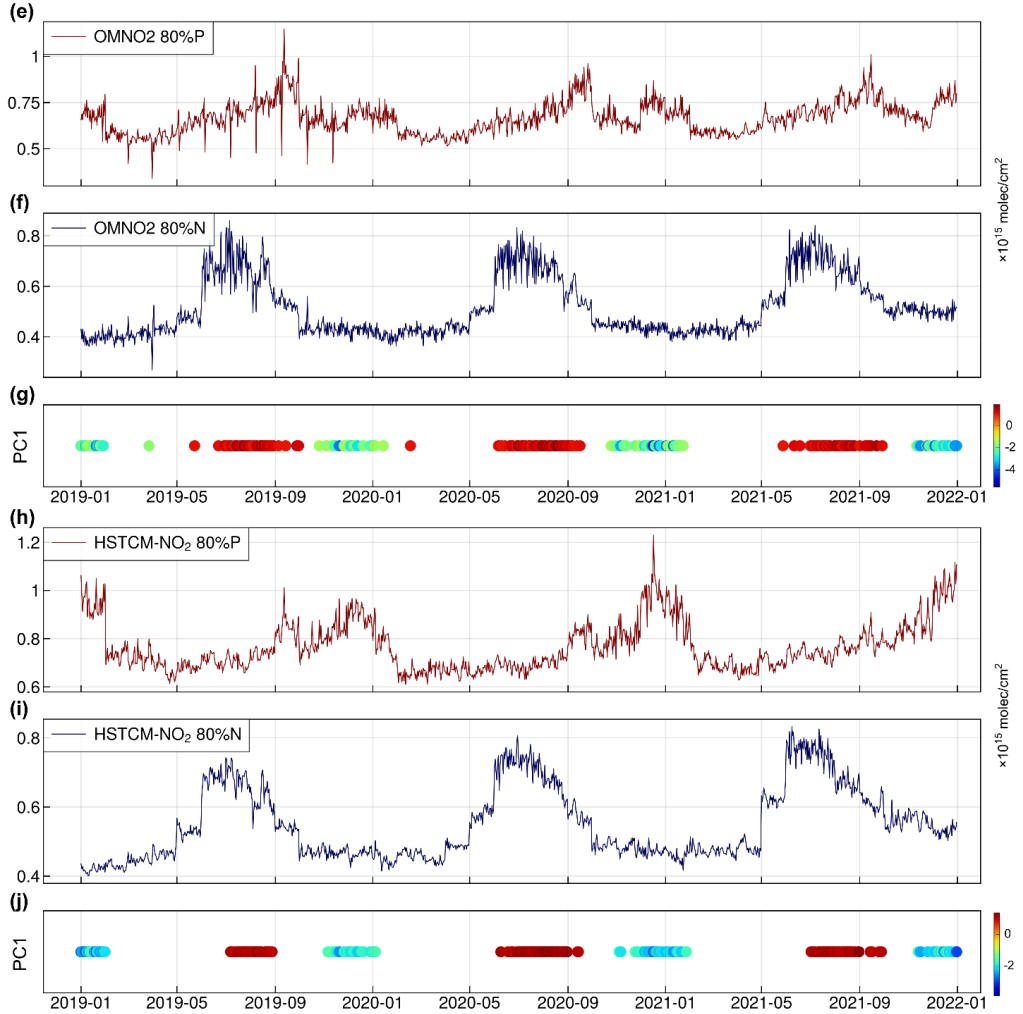


**Figure 12:** Spatial and temporal patterns after EOF variance maximization is performed on both OMNO2 and HSTCM-NO₂. EOF1 is given for OMNO2 positive **(a)**, OMNO2 negative **(b)**, HSTCM-NO₂ positive **(c)**, and HSTCM-NO₂ negative **(d)**. The temporal mean value of OMNO2 over the EOF1 positive region and EOF1 negative region are respectively given in **(e)** and **(f)**, while PC1 is given in **(g)**, where red and blue represent the peaks in the positive and negative factors respectively. Subfigures **(h)**, **(i)**, and **(j)** are similar to **(e)**, **(f)**, and **(g)** except
when applied to HSTCM-NO₂.

Figure 12 shows the spatial and temporal patterns after EOF variance maximization is performed on both OMNO2 and HSTCM-NO₂. First and foremost, the EOFs represent a few general patterns, seeming to capture a combination of biomass burning (across Africa, South America, and Australia), urbanization (across South Africa, Northeastern China and Japan), energy producing regions in the Southern US and northern Mexico, and transport regions from the Mediterranean to the Indian
Ocean. This includes the large areas of pollution transported downwind over the various oceans, and uncertainty associated with clouds, sea salt, and low signal strengths near where the Southern Ocean intrudes into the cleaner areas of the Indian and Pacific Oceans respectively. The overall patterns look reasonable in both space and time.

A more details analysis clearly demonstrates that three such examples are consistently represented between the original OMNO2 and HSTCM-NO₂. First, the negative mode of EOF1 representing biomass burning over Congo and its subsequent
transport over the Southern Atlantic Ocean, and the positive mode of EOF1 representing biomass burning and urbanization over respective parts of Southern Africa are interpolated well and line-filled by the respective negative and positive modes of



HSTCM-$NO_2$ EOF1 (Du et al., 2020). Second, the wildfires off of Southwestern Australia and subsequent transport into the Southern Ocean are clearly shown by the negative mode of EOF1, while the negative mode of EOF1 of HSTCM-$NO_2$ expands these observations into the Indian Ocean and all the way to New Zealand, while narrowing the band and reducing the error due to the mixing from the Southern Ocean, consistent with observations (Wenig et al., 2003). Third, the positive region of EOF1 loosely picks up the transported wave-trains from East Asia to North America, while the HSTCM-$NO_2$ is able to clearly pick up the entire wave-train clearly originating in industrial regions of Japan and spreading part of the time to Luzon and another part of the time to the USA (Wang, Ma, et al., 2023). In terms of time, it is clear that the negative EOF1 regions in both plots are well represented by the positive PC1 values. All three peaks demonstrated are clearly observed in the average values of $NO_2$ over the negative EOF1 regions respectively. There are 4 large peaks and two small peaks represented in the negative PC1 values, all of which are picked up well in the average values of $NO_2$ over the positive EOF1 regions respectively. All of the peak times are represented in the time series using different colors.

While the spatial distribution of the HSTCM-$NO_2$ EOF is more smeared spatially than the OMNO2 product in some regions, this is not unexpected. In some cases, this makes the story consistent, by filling in missing data, especially so in cases of long-range transported plums which are otherwise missing, as well as for the known variation observed over Henan and Shandong. However, some of the smearing is also noise, as identified over the low $NO_2$ concentration regions near where the Indian and Pacific Oceans intersect with the Southern Ocean.

This analysis shows that the HSTCM-$NO_2$ product does a decent job at representing the temporal and spatial extremes in the original OMNO2 dataset. While this test is not frequently done in the community (Cohen, 2014; Liu et al., 2023; Liu et al., 2024), it clearly demonstrates in an objective manner a new and additional way to test the goodness of the final product, in that it requires it to not only match in space and time with observed mean conditions, but also with observed extreme conditions. The fact that there is spatial smearing in some aspects is good, in that it fills in missing long-range transport events that are missed between swaths or due to clouds in-situ. In other aspects, it may extend the actual signals too far in space. For these reasons, care must be used when applying the results. We hope that this section sets a gold standard by which future big data products are more carefully compared with and validated against the underlying data.

## 4 Data availability

The global daily high spatial-temporal coverage merged tropospheric $NO_2$ dataset (HSTCM-$NO_2$) from 2007 to 2022 based on OMI and GOME-2 can be accessed directly through: https://doi.org/10.5281/zenodo.10968462 (Qin et al., 2024).

## 5 Conclusions and discussion

In order to improve the spatial coverage of OMNO2 due to data loss caused by cloud occlusion, row anomaly, high retrieval noise, and other issues, this study proposes an effective method of reconstruction consisting of machine learning (XGBoost) and gap filling (DINEOF) to produce a new reconstructed product (HSTCM-$NO_2$).

First, the process of applying XGBoost first followed by DINEOF second yields the highest correlation and lowest RMSE between the OMNO2 and HSTCM-$NO_2$. In specific, the process of first applying XGBoost, and then following with DINEOF is found to be most efficient. One reason for this is that XGBoost requires the presence of GOME-2 data, allowing for additional observational support in the final reconstructed product. This is consistent with the fact that GOME-2 occupies a very high SHAP value. There are a few qualifiers however: first that cases without prior knowledge perform less well than places with priori knowledge; and second that locations with a lower column loading of OMNO2 work better than places with a higher column loading of OMNO2. Since the majority of the data points globally are biased towards lower (i.e. non-polluted) areas, comparison with additional datasets and using different approaches is essential.

Second, external observations from MAX-DOAS and TROPOMI as well as reanalysis data from EAC4 are used to validate



HSTCM-NO$_2$ on a column-by-column, large-area basis. HSTCM-NO$_2$ shows good correlation with all of the observations above, especially so when the VCDs are below $6\times10^{15}$ molec.cm$^{-2}$. Specific issues in terms of spatial distribution mis-matches and issues reproducing very high VCDs are explained in detail within the paper. There are a few exceptions to this, specifically over Wuhan and the Yangtze River from Wuhan up to Nanjing, and specific urban parts of India (such as New Delhi) being reasonably well represented.

Third, additional analysis to verify the goodness of HSTCM-NO$_2$ in terms of being able to capture extreme events observed within the OMNO2 data is also performed. In this case, variance maximization is used to decompose the OMNO2 data into standing spatial (EOF) and temporal (PC) signals. A similar analysis is performed on the HSTCM-NO$_2$ data, with the resulting signals compared. It is shown that in addition to generally matching in terms of space and time, that missing data due to gap filling observed by HSTCM-NO$_2$ especially downwind from large pollution areas over various oceans (South Atlantic, Indian, South Pacific and North Pacific) are improved. Interestingly, some of the strongest signals, including biomass burning from central and northern Africa including in Algeria, including pixels over the value of $6\times10^{15}$ molec.cm$^{-2}$ are also well represented, in terms of both the magnitude, as well as the spatial and temporal extremes.

This combination of findings indicates that the new HSTCM-NO$_2$ product works well in terms of representing both the grid-by-grid and climatological mean conditions, as well as extreme events, with the caveats that first there is some a priori knowledge and second that the original OMNO2 data has an VCD below $6\times10^{15}$ molec.cm$^{-2}$ (i.e. is not heavily polluted).

## 6 Competing interests

The contact author has declared that none of the authors has any competing interests.

## 7 Financial support

This research has been supported by the National Natural Science Foundation of China (Grant No. 42375125).

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
