# Peer review of "A Global Daily High Spatial-temporal Coverage Merged Tropospheric NO2 dataset (HSTCM-NO2) from 2007 to 2022 based on OMI and GOME-2"

_Earth System Science Data, 2024_

## Author Comment (AC1)

To facilitate the response, all unaltered original comments are highlighted in yellow while our responses and revisions are highlighted in blue.

**Reviewer 1:**

1. In the abstract, they should introduce the model performance, such as the cross validation and external validation results.

   Model performance metrics have been clearly included and are ow expanded upon in section 3.2. The current version has spent time to not only validate using traditional techniques, but also a new best of class technique in terms of matching the EOFs with the underlying data, as explained in section 2.6.
   Please let us know if this must still be improved.

2. Table 1. I think the figures in the table are not necessary. Please delete them to make the table more concise.

   We have deleted the figures in Table 1 to make it more concise.

3. What is the purpose of Lines 108-112? It seems not relevant to the sections 2.1.1-2.1.3.

   The original Lines 108-112 have been deleted.

4. Section 2.5 should be simplified. There is no need to provide the equations of R2, RMSE, etc. Most people know them.

   Equations in original Section 2.5 have been deleted, now we only mention which metrics are used in the article.

5. Delete "2.6 Empirical Orthogonal Functions" and change 2.7 to 2.6.

   Thank you for catching this error. This has already been deleted, and the number of sections has been rearranged and checked.

6. The validation method is not clear. I suggest them adding a section to introduce their validation strategy, including cross-validation and external validation using MAX-DOAS, other satellites (TROPOMI), and reanalysis products.

   A new section 2.6 "Validation strategy" has been added. This section describes our validation strategy in detail, and incorporates an elaboration of the original metrics as pointed out by the reviewer. This section also includes some sentences on how we perform EOF analysis on the observed and gap-filled data, and why this is an

important additional form of validation that we hope the community will start to use (from Line 204 to Line 209): "Also, as an important and innovative approach, EOF is performed on the three-dimensional observed and HSTCM-$NO_2$ fields. These values are compared to ensure that the most significant changes in the spatial and temporal pattern are consistent with the same most significant changes in the spatial and temporal pattern of the original observations. EOF is an exploratory technique for multivariate data, reducing the data to an eigenvalue problem that explains and interprets the variability in the data. EOF has been used in other studies of data analysis using satellite-based remote sensing to estimate the spatiotemporal distribution and characteristics of pollutants including: HCHO (Kim et al., 2014), CO (Baek & Kim, 2011), aerosols (Cohen et al., 2017) and $NO_2$ (Li et al., 2023)."

7. Figure 9, please add the time scope.

We have enhanced the annotation of time in the figure and clarified the time scope in both the text and title.

8. Some previous studies have also fill OMI $NO_2$ gaps in some countries such as in China. Please introduce them in the introduction section if necessary. e.g., Shao et al, 2023, Estimation of daily $NO_2$ with explainable machine learning model in China, 2007-2020; Wu et al, 2023, A robust approach to deriving long-term daily surface $NO_2$ levels across China: Correction to substantial estimation bias in back-extrapolation.

We think this part is necessary to increase the number of relevant studies cited, as a means to enhance the introduction. We added some references in Lines 82-84: "Due to 19 years of continuous observations, OMI is a very widely used sensor in the field of atmospheric trace gas research, and finding ways to comprehensively and reasonably fill these missing pixels would allow its usefulness to be extended into other fields (de Hoogh et al., 2019; He et al., 2020; Wu et al., 2021; Wei et al., 2022; Shao et al., 2023; Liu et al., 2024)". Specific references added are:
- de Hoogh, K., Saucy, A., Shtein, A., Schwartz, J., West, E. A., Strassmann, A., Puhan, M., Röösli, M., Stafoggia, M., and Kloog, I.: Predicting Fine-Scale Daily $NO_2$ for 2005–2016 Incorporating OMI Satellite Data Across Switzerland, Environ. Sci. Technol., 53, 10279–10287, https://doi.org/10.1021/acs.est.9b03107, 2019.
- Wu, Y., Di, B., Luo, Y., Grieneisen, M. L., Zeng, W., Zhang, S., Deng, X., Tang, Y., Shi, G., Yang, F., and Zhan, Y.: A robust approach to deriving long-term daily surface $NO_2$ levels across China: Correction to substantial estimation bias in back-extrapolation, Environ. Int., 154, 106576, https://doi.org/10.1016/j.envint.2021.106576, 2021.
- Shao, Y., Zhao, W., Liu, R., Yang, J., Liu, M., Fang, W., Hu, L., Adams, M., Bi, J., and Ma, Z.: Estimation of daily $NO_2$ with explainable machine learning model in China, 2007–2020, Atmos. Environ., 314, 120111, https://doi.org/10.1016/j.atmosenv.2023.120111, 2023.

- Liu, J., Cohen, J. B., He, Q., Tiwari, P., and Qin, K.: Accounting for NOx emissions from biomass burning and urbanization doubles existing inventories over South, Southeast and East Asia, Commun. Earth Environ., 5, 255, https://doi.org/10.1038/s43247024014245, 2024.

9. HSTCM-NO$_2$ can improve the data to full coverage. This should be mentioned in abstract. Besides, "which increases the global spatial coverage of NO$_2$ by ~60% compared to the original OMINO2 data", the 60% here has ambiguity. I believe 60% here is the absolute coverage. But it can be misunderstood as the 60% of the original OMI data. Also revise relevant statement in the main text.

This increase of 60% is with respect to the entire global coverage. This is now mentioned more clearly by emphasizing the concept of "spatial coverage". And the statement in the abstract has been adjusted to "…which increases the average global spatial coverage of NO$_2$ from 39.5% to 99.1%".

10. The method of SHAP should be moved to the method section.

"2.4 SHAP (SHapley Additive exPlanation) values" is used to explain the method of SHAP.

**Reviewer 2/3:**

1. As the study combines satellite data with morning and afternoon overpass time, additional recommendations for data use, such as data assimilation and model comparison, are suggested.

Thank you for helping us to explain the work more precisely and in detail. We have added additional details to the preprocessing part, specifically outlining how data has been harmonized (see Section 2.1). The issue of how to address temporal difference in terms of assimilation of satellite data is a much harder problem that deserves further in-depth exploration. Finding better models for reconstruction in this manner looks like an interesting area for future study. Thank you for your valuable suggestions!

2. Second, polluted scenes are typically drawing more attention and performing less well in this work, therefore comments on how to improve the data for such scenes are recommended.

Our separation of the data into surface and ocean has helped to some extent. However, this is both related to the retrieval itself as well as pollution levels. The fact that the EOF is able to capture known biomass burning plumes and their transport also shows that there is some improved ability to track polluted events,

even those which cross the land/sea boarder as implemented herein. However, the data also points clearly to an issue requiring additional work at both medium and high (>6×10$^{15}$ molec.cm$^{-2}$) NO$_2$ levels.

We have added the following at the end of the discussion:

"In the future, related work will focus on how to enhance the application of datasets in polluted scenes. Separating low and high values for training might be an effective approach, since it is known that there are different retrieval assumptions and impacts that occur under polluted and non-polluted conditions (Boersma et al., 2007; Chimot et al., 2016; Lorente et al., 2018; Liu et al., 2019; Zhou et al., 2024). Presently the criteria for demarcation and the sets of impacting variables are still undergoing discussion by the community and are not yet agreed upon. Further consideration needs to be made whether there are better methods or combinations of methods that can be applied across the full range of scenarios at the same time".

3. In addition, titles, dates, and/or colorbars in some figures are difficult to recognize. Please enlarge them for maps.

We examined the figures and adjusted the font size in Figures 4 and 9. Thank you for helping us make the results easier to follow.

4. Content of Sect. 2.6 is missing.

The section on EOF in the original text has now been supplemented and the subsections have been reorganized. Please see the response to reviewer 1 above.

5. Line 93 What are the advantages of machine learning and pattern recognition specifically? How do you compare the methods and results to previous works in terms of consistency and difference?

We believe that the methodology used in this paper is one that is able to take into account both spatial geographic correlation and inter-sensor correlation, and is applicable to large datasets, thus differentiating it from previously existing studies, as already expressed in the article: "As there is strong correlation in terms of both geospatial relationships as well as retrieval approaches used to determine the VCDs between tropospheric NO$_2$ obtained by different sensors (Wang et al., 2016; Park et al., 2020), issues of spatial-temporal correlation also needs to be carefully taken into consideration, something that previous studies may not have fully considered".

We also believe that this approach is different from some other approaches, in that we have not considered the impacts of observational uncertainty on the machine learning itself, which is a very new yet important finding (He et al., 2024).

- He, Q., Qin, K., Cohen, J. B., Li, D., & Kim, J. (2024). Quantifying Uncertainty in ML-derived Atmosphere Remote Sensing: Hourly Surface NO$_2$ Estimation with GEMS. Geophysical Research Letters. e2024GL110468. (Accepted)

6. Line 131 How do you combine the three datasets and deal with their differences in instrument and algorithm?

We have added clarification here: "Datasets were resampled at uniform gridding of 0.25×0.25 degree using the HARP tool".

7. Line 156 What is the reason to select these 3 stations?

We added the reason for using these 3 sites in Lines 158-159: "The sites are categorized into three types (Sub-urban, Urban and Rural) based on their location, with each of the three sites having a different use-type applied herein".

8. Line 219 was -> is

Thank you for helping us to identify this omission, which has been corrected.

9. Line 230 The slope and intercept deserve some discussion, as method 1 shows a reduced performance.

We believe that using a combination of both the R and the RMSE variables, that method 1 performs at least as well as method 2 and method 3. We also demonstrate that across all of the methods, including GOME-2 observations increases the overall accuracy of reconstruction. We do agree that the slope and intercept may not be very high, but this is due to the very large amount of data less than $6\times10^{15}$ molecules/cm$^2$, as talked about further into the work. The fact that the big data model using method 1 is able to capture both the spatial and temporal variability, including of extreme events within the lower concentration range (less than $6\times10^{15}$ molecules/cm$^2$), adding further support to the fit being reasonable over most of the globe.

While the overall fitting values are not as high as some very idealized case studies, the fact that the filled in data represents more than half of the global pixels, it is not expected that the $R^2$ should be too high or the fits should be very perfect. In fact, in such a case, this would result in overfitting, leading to extreme matters patterns observed at high spatial and temporal frequency being not well represented (He et al., 2024).

We fleshed out the original expression: "Meanwhile, by comparing Column 2 and Column 3, it is obvious that the presence of GOME-2 observations can greatly improve the accuracy of reconstruction and have an impact on the fitted slopes (especially in the case of methods I and II)".

10. Line 244 1952 -> 1952)

This error has been fixed and the related part is now in Section 2.4.

11. Line 310 .,->.

This has been corrected.

12. Line 325 VCD and vertical column concentration are used Interchangeably, better be consistent. Please also be consistent with MAXDOAS or MAX-DOAS, machine learning or machine-learning, etc.

These three terms have been harmonized as "VCD", "MAX-DOAS" and "machine learning" respectively.

13. Line 394 Which color shows the results using both XGBoost and DINEOF? What does the red line show in the figure?

The sum of the points marked by all colors is the result of the comparison between the final reconstructed product and MAX-DOAS, both the red line and the overall result are located in the upper left corner of the figure box, which we have added a related narrative – "The boxes in the upper left corner summarize the statistical comparisons, while the boxes to the right of each subfigure represent the statistics of each individual reconstruction step" in the figure name to illustrate. Also, we have added the purple y=x baseline.

14. Line 420 Define the abbreviation RA first.

We have added it where it was first mentioned in the paper (Line 78).

---

## Author Response (AR1)

Dear Editor and Reviewers,

We appreciate all of the time you have provided. Your insightful and valuable comments and recommendations have helped us make this work stronger. We believe the responses below demonstrate an improved version of the manuscript and data, and have addressed all of your concerns and queries. We have further carefully re-read and proofed the entire manuscript.

To facilitate the response, your unaltered original remarks are highlighted in yellow while our responses and revisions are highlighted in blue.

We believe that our revised submission should meet the rigorous standards of Earth System Science Data, and want to thank you for your consideration. If you have any further questions or comments, please let us know.

Best Regards,

Jason Blake Cohen (On Behalf of the Authors)
jasonbc@alum.mit.edu
School of Environment and Spatial Informatics
China University of Mining and Technology (CUMT)

*Reviewer 1:*

1. In the abstract, they should introduce the model performance, such as the cross validation and external validation results.

Model performance metrics have been clearly included and expanded in Section 3.2. The current version has spent time to both do validation using traditional techniques, as well as introduced a new best of class validation technique in terms of matching the EOFs of the results with the underlying data, as explained in Section 2.6.

2. Table 1. I think the figures in the table are not necessary. Please delete them to make the table more concise.

We have deleted the figures in Table 1 to make it more concise.

3. What is the purpose of Lines 108-112? It seems not relevant to the sections 2.1.1-2.1.3.

The original Lines 108-112 have been deleted.

4. Section 2.5 should be simplified. There is no need to provide the equations of R2, RMSE, etc. Most people know them.

Equations in the original Section 2.5 have been deleted, now we only mention which metrics are used in the article.

5. Delete "2.6 Empirical Orthogonal Functions" and change 2.7 to 2.6.

Thank you for catching this error. This has been deleted, and the number of the sections has been rearranged and checked.

6. The validation method is not clear. I suggest them adding a section to introduce their validation strategy, including cross-validation and external validation using MAX-DOAS, other satellites (TROPOMI), and reanalysis products.

A new section 2.6 "Validation strategy" has been written. This section describes our validation strategy in detail, and incorporates an elaboration of the original metrics as pointed out by the reviewer. This section also includes sentences on how we perform EOF analysis on the observed and gap-filled data, and why this is an important additional form of validation that we hope the community will start to also use (from Line 206 to Line 211):

"Also, as an important and innovative approach, EOF is performed on the three-dimensional HSTCM-$NO_2$ fields and compared against the EOF patterns applied to the observations, to ensure that the maximum changes in spatial and temporal signal are consistent with the original observations. EOF is an exploratory technique for multivariate data, which is in essence an eigenvalue problem, aiming at explaining and interpreting the variability in the data. Till now, EOF has been introduced into data analysis of satellite-based remote sensing to estimate the spatiotemporal distribution characteristics of pollutants such as HCHO (Kim et al., 2014), CO (Baek & Kim, 2011), aerosols (Cohen et al., 2017) and $NO_2$ (Li et al., 2023)."

7. Figure 9, please add the time scope.

We have both enhanced the annotation of time in the figure and clarified the time scope in both the text and title.

8. Some previous studies have also fill OMI $NO_2$ gaps in some countries such as in China. Please introduce them in the introduction section if necessary. e.g., Shao et al, 2023, Estimation of daily $NO_2$ with explainable machine learning model in China, 2007-2020; Wu et al, 2023, A robust approach to deriving long-term daily surface $NO_2$ levels across China: Correction to substantial estimation bias in back-extrapolation.

We have increased the number of relevant studies cited to deepen the impact of the introduction. We added some references in Lines 81-84:
"Due to 19 years of continuous observations, OMI is a very widely used sensor in the field of atmospheric trace gas research, and finding ways to comprehensively and reasonably fill these missing pixels would allow its usefulness to be extended into other fields (de Hoogh et al., 2019; He et al., 2020; Wu et al., 2021; Wei et al., 2022; Shao et al., 2023; Liu et al., 2024)".
The specific references added include:
- de Hoogh, K., Saucy, A., Shtein, A., Schwartz, J., West, E. A., Strassmann, A., Puhan, M., Röösli, M., Stafoggia, M., and Kloog, I.: Predicting Fine-Scale Daily $NO_2$ for 2005–2016 Incorporating OMI Satellite Data Across Switzerland, Environ. Sci. Technol., 53, 10279–10287, https://doi.org/10.1021/acs.est.9b03107, 2019.

- Wu, Y., Di, B., Luo, Y., Grieneisen, M. L., Zeng, W., Zhang, S., Deng, X., Tang, Y., Shi, G., Yang, F., and Zhan, Y.: A robust approach to deriving long-term daily surface $NO_2$ levels across China: Correction to substantial estimation bias in back-extrapolation, Environ. Int., 154, 106576, https://doi.org/10.1016/j.envint.2021.106576, 2021.
- Shao, Y., Zhao, W., Liu, R., Yang, J., Liu, M., Fang, W., Hu, L., Adams, M., Bi, J., and Ma, Z.: Estimation of daily $NO_2$ with explainable machine learning model in China, 2007–2020, Atmos. Environ., 314, 120111, https://doi.org/10.1016/j.atmosenv.2023.120111, 2023.
- Liu, J., Cohen, J. B., He, Q., Tiwari, P., and Qin, K.: Accounting for NOx emissions from biomass burning and urbanization doubles existing inventories over South, Southeast and East Asia, Commun. Earth Environ., 5, 255, https://doi.org/10.1038/s43247024014245, 2024.

9. HSTCM-$NO_2$ can improve the data to full coverage. This should be mentioned in abstract. Besides, "which increases the global spatial coverage of $NO_2$ by ~60% compared to the original OMINO2 data", the 60% here has ambiguity. I believe 60% here is the absolute coverage. But it can be misunderstood as the 60% of the original OMI data. Also revise relevant statement in the main text.

This increase of 60% is with respect to the entire global coverage. This is now mentioned more clearly by emphasizing the concept of "spatial coverage". The statement in the abstract has been adjusted to "…which increases the average global spatial coverage of $NO_2$ from 39.5% to 99.1%".

10. The method of SHAP should be moved to the method section.

"2.4 SHAP (SHapley Additive exPlanation) values" is now included in the paper.

**Reviewer 2/3:**

1. As the study combines satellite data with morning and afternoon overpass time, additional recommendations for data use, such as data assimilation and model comparison, are suggested.

Thank you for helping us to explain the work more precisely and in detail. We have added additional details to the preprocessing part, specifically outlining how data has been harmonized (see Section 2.1). The issue of how to address temporal difference in terms of assimilation of satellite data is a much harder problem that deserves further in-depth exploration. Finding better models for reconstruction in this manner looks like an interesting area for future study. Thank you for your valuable suggestions!

2. Second, polluted scenes are typically drawing more attention and performing less well in this work, therefore comments on how to improve the data for such scenes are recommended.

Our separation of the data into surface and ocean has helped to some extent. However, this is both related to the retrieval itself as well as pollution levels. The fact that the EOF is able to capture known biomass burning plumes and their transport also shows that there is some improved ability to track polluted events, even those which cross the land/sea boarder we have implemented herein. However, the data also points clearly requiring additional work at both medium and high ($>6\times10^{15}$ molec.cm$^{-2}$) loadings.

We have added the following at the end of the discussion:
"In the future, related work will focus on how to enhance the application of datasets in polluted scenes. Separating low and high values for training might be an effective approach, since it is known that there are different retrieval assumptions and impacts that occur under polluted and non-polluted conditions (Boersma et al., 2007; Chimot et al., 2016; Lorente et al., 2018; Liu et al., 2019; Zhou et al., 2024). Presently the criteria for demarcation and the sets of impacting variables are still undergoing discussion by the community and are not yet agreed upon. Whether there are better methods or combinations of methods that can be applied across the full range of scenarios at the same time is also something that needs to be considered".

3. In addition, titles, dates, and/or colorbars in some figures are difficult to recognize. Please enlarge them for maps.

We examined the figures and adjusted the font size in Figures 4 and 9. Thank you for helping us make the results easier to follow.

4. Content of Sect. 2.6 is missing.

The section on EOF in the original text has now been supplemented and the subsections have been reorganized.

Please see the response to reviewer 1 above.

5. Line 93 What are the advantages of machine learning and pattern recognition specifically? How do you compare the methods and results to previous works in terms of consistency and difference?

We believe that the methodology used in this paper is one that is able to take into account both spatial geographic correlation and inter-sensor correlation, and is applicable to large datasets, thus differentiating it from previously existing studies, as already expressed in the article: "As there is strong correlation in terms of both geospatial relationships as well as retrieval approaches used to determine the VCDs between tropospheric $NO_2$ obtained by different sensors (Wang et al., 2016; Park et al., 2020), issues of spatial-temporal correlation need to be carefully taken into consideration, something that these previous approaches may not have fully considered".

We also believe that this approach is different from some other approaches, in that we have not considered the impacts of observational uncertainty on the machine learning itself, which is a very new yet important finding (He et al., 2024).

- He, Q., Qin, K., Cohen, J. B., Li, D., & Kim, J. (2024). Quantifying Uncertainty in ML-derived Atmosphere Remote Sensing: Hourly Surface $NO_2$ Estimation with GEMS. Geophysical Research Letters. e2024GL110468. (Accepted)

6. Line 131 How do you combine the three datasets and deal with their differences in instrument and algorithm?

We have added clarification here: "Datasets were resampled at uniform gridding of 0.25×0.25 degree using HARP".

7. Line 156 What is the reason to select these 3 stations?

We added the reason for using these 3 sites in Lines 158-159: "The sites are categorized into three types (Sub-urban, Urban and Rural) based on their location, and 3 sites of different types are used here".

8. Line 219 was -> is

Thank you for helping us to identify this omission, which has been corrected.

9. Line 230 The slope and intercept deserve some discussion, as method 1 shows a reduced performance.

We believe that using a combination of both the R and the RMSE statistics, that method 1 performs at least as well as method 2 and method 3. We also demonstrate that across all of the methods, including GOME-2 observations increases the overall accuracy of reconstruction. We do agree that the slope and intercept may not be very high, but this is due to the very large amount of data less than $6\times10^{15}$ molec.cm$^{-2}$, as talked about later in the work. The fact that the big data model using method 1 is able to capture both the spatial and temporal variability, including extreme events within the lower concentration range (i.e. when the $NO_2$ concentration is lower than $6\times10^{15}$ molec.cm$^{-2}$) as discussed in Section 3.3 and Section 5, adds further support to the fit being reasonable over most of the globe.

While the overall fitting values are not as high as some very idealized case studies, the fact that the filled in data represents more than half of the global pixels, it is not expected that the $R^2$ should be too high or the fits should be very perfect. In fact, if this were the case, it likely would be overfitting, leading to a case where actual patterns observed at high spatial and temporal frequency are not well represented (He et al., 2024).

- He, Q., Qin, K., Cohen, J. B., Li, D., & Kim, J. (2024). Quantifying Uncertainty in ML-derived Atmosphere Remote Sensing: Hourly Surface $NO_2$ Estimation with GEMS. Geophysical Research Letters. e2024GL110468. (Accepted)

We fleshed out the original expression: "Meanwhile, by comparing Column 2 and Column 3, it is obvious that the presence of GOME-2 observations can greatly improve the accuracy of reconstruction and have an impact on the fitted slopes (especially in the cases of methods I and II)".

10. Line 244 1952 -> 1952)

This error has been fixed and the related part is now located at Section 2.4.

11. Line 310 .,->.

This has been corrected.

12. Line 325 VCD and vertical column concentration are used Interchangeably, better be consistent. Please also be consistent with MAXDOAS or MAX-DOAS, machine learning or machine-learning, etc.

These three terms have been harmonized as "VCD", "MAX-DOAS" and "machine learning" respectively.

13. Line 394 Which color shows the results using both XGBoost and DINEOF? What does the red line show in the figure?

The sum of the points marked by all colors is the result of the comparison between the final reconstructed product and MAX-DOAS, both the red line and the overall result are located in the upper left corner of the figure box, which we have added a related narrative – "The boxes in the upper left corner summarize the statistical comparisons, while the boxes to the right of each subfigure represent the statistics of each individual reconstruction step" in the figure name to illustrate. Also, we have added the purple y=x baseline.

14. Line 420 Define the abbreviation RA first.

We have added it where it was first mentioned in the paper (Line 78).

[revised manuscript text omitted]